# Spatial and temporal intratumour heterogeneity has potential consequences for single biopsy-based neuroblastoma treatment decisions

Karin Schmelz[1,2,3,13], Joern Toedling [1,2,3,13], Matt Huska [4,13], Maja C. Cwikla [1,4], Louisa-Marie Kruetzfeldt [1], Jutta Proba[1], Peter F. Ambros[5], Inge M. Ambros[5], Sengül Boral[1], Marco Lodrini[1,2,3], Celine Y. Chen [1,6], Martin Burkert[4], Dennis Guergen[7], Annabell Szymansky [1], Kathy Astrahantseff [1], Annette Kuenkele[1,2,3,8], Kerstin Haase [1,2,3,8], Matthias Fischer [9,10], Hedwig E. Deubzer [1,2,3,6,8], Falk Hertwig [1,2,3], Patrick Hundsdoerfer[1,11], Anton G. Henssen [1,2,3,4,6,8,14✉], Roland F. Schwarz [4,12,14✉], Johannes H. Schulte [1,2,3,8,14✉] & Angelika Eggert [1,2,3,8,14✉]

Intratumour heterogeneity is a major cause of treatment failure in cancer. We present in-depth analyses combining transcriptomic and genomic profiling with ultra-deep targeted sequencing of multiregional biopsies in 10 patients with neuroblastoma, a devastating childhood tumour. We observe high spatial and temporal heterogeneity in somatic mutations and somatic copy-number alterations which are reflected on the transcriptomic level. Mutations in some druggable target genes including *ALK* and *FGFR1* are heterogeneous at diagnosis and/or relapse, raising the issue whether current target prioritization and molecular risk stratification procedures in single biopsies are sufficiently reliable for therapy decisions. The genetic heterogeneity in gene mutations and chromosome aberrations observed in deep analyses from patient courses suggest clonal evolution before treatment and under treatment pressure, and support early emergence of metastatic clones and ongoing chromosomal instability during disease evolution. We report continuous clonal evolution on mutational and copy number levels in neuroblastoma, and detail its implications for therapy selection, risk stratification and therapy resistance.

[1] Charité—Universitätsmedizin Berlin, Berlin, Germany. [2] The German Cancer Consortium (DKTK), Partner Site Berlin, Berlin, Germany. [3] The German Cancer Research Center (DKFZ), Heidelberg, Germany. [4] Berlin Institute for Medical Systems Biology (BIMSB), Max Delbrück Center for Molecular Medicine in the Helmholtz Association (MDC), Berlin, Germany. [5] Children's Cancer Research Institute, St. Anna Kinderkrebsforschung, 1090 Vienna, Austria. [6] Experimental and Clinical Research Center (ECRC) of the Charité and Max Delbrück Center for Molecular Medicine in the Helmholtz Association, Berlin, Germany. [7] Experimental Pharmacology and Oncology Berlin-Buch GmbH (EPO), Berlin, Germany. [8] Berlin Institute of Health (BIH), Berlin, Germany. [9] Department of Experimental Pediatric Oncology, Medical Faculty, University Children's Hospital of Cologne, Cologne, Germany. [10] Center for Molecular Medicine Cologne (CMMC), University of Cologne, Cologne, Germany. [11] Helios Klinikum Berlin-Buch, Berlin, Germany. [12] BIFOLD—Berlin Institute for the Foundations of Learning and Data, Berlin, Germany. [13] These authors contributed equally: Karin Schmelz, Joern Toedling, Matt Huska. [14] These authors jointly supervised this work: Anton G. Henssen, Roland F. Schwarz, Johannes H. Schulte, Angelika Eggert. ✉email: anton.henssen@charite.de; roland.schwarz@mdc-berlin.de; johannes.schulte@charite.de; angelika.eggert@charite.de

Neuroblastoma is the most common solid paediatric tumour, accounting for 15% of cancer-related deaths in early childhood. Especially patients directly diagnosed with high-risk neuroblastoma have a poor prognosis[1,2]. High-risk (HR) and ultra-high-risk (UHR) neuroblastoma are characterised by active telomere maintenance and mutations in distinct pathways defining specific clinically relevant molecular risk classifiers[3]. Telomere maintenance as the hallmark of the UHR concept is either conveyed by *TERT* gene activation due to *TERT* promoter mutations, *TERT* re-arrangements or high *MYCN* expression or by alternative lengthening of telomeres (ALT)[4]. Intriguingly, even high-risk neuroblastomas initially respond to treatment, but eventually relapse as chemotherapy-resistant disease. Current target prioritisation efforts in relapsed/refractory patients such as INFORM, MAPPYACTS, NCI-COG MATCH, PROFYLE or ITHER[5–8] rely on molecular profiling based on bulk sequencing approaches in single tumour biopsies. Targeted therapy based on single-biopsy profiles is often inefficient or only exhibits short-term effects, and rarely results in long-term tumour control. Secondary resistance following promising short-term efficacy is clinically observed in particular for activating *ALK* mutations, the most recurrent actionable single-nucleotide variant (SNV) present in neuroblastoma for which targeted inhibitors of 1st, 2nd and now 3rd generation have entered clinical trials[9–12]. These clinical observations indicate the existence of therapy-resistant subclones in high-risk neuroblastoma. Indeed, neuroblastoma shows extensive genetic intratumour heterogeneity and distinct evolutionary patterns[13–15]. Both linear/late-branching as well as parallel/early-branching evolution occurs in neuroblastoma, and impacts clinical behaviour differently, with early branching being associated with adverse outcome[14]. Thus, intratumour heterogeneity is an important feature of neuroblastoma, which deserves in-depth analysis and further interpretation with respect to potential clinical implications.

While neuroblastoma typically has few SNVs, neuroblastoma genomes show pronounced somatic copy-number alterations (SCNA)[16]. Chromosomal instability describes the progressive accumulation of SCNAs over time, equating to changes in the number and structure of chromosomes during tumour evolution. Chromosomal instability results from chromosome segregation errors during cell division, and high levels have been linked to poor prognosis[17,18] and multi-drug resistance[19,20] in many solid tumours. Neuroblastoma is widely considered to be driven by chromosomal instability since high levels of multiple clinically relevant SCNA events occur with a low overall mutational burden. Focal amplifications of the *MYCN* oncogene, re-arrangements within the *TERT* locus on chromosome 5p and SCNAs on chromosomes 1p36, 11q and 17q, genetically define a high-risk clinical phenotype[4,21,22]. Despite the clinical relevance of chromosomal instability in neuroblastoma and recent evidence in other cancers that ongoing chromosomal instability keeps shaping cancer genomes during disease progression[23–25], the extent of ongoing chromosomal instability after initial biopsy at diagnosis and the dynamics of SCNA acquisition during neuroblastoma progression and treatment remain unclear.

To address open questions about neuroblastoma intratumour heterogeneity, we combine multi-region transcriptome and whole-exome sequencing, followed by ultra-deep targeted sequencing of 140 spatially and temporally separated tumour samples from 10 clinically heterogeneous patients with neuroblastoma. We describe spatial and temporal intratumour heterogeneity of actionable mutations and distinct patterns of tumour evolution based on SNVs and SCNAs.

## Results

### Analysis of SNVs in spatially and temporally distinct neuroblastoma samples reveals extensive intratumour heterogeneity.

To analyse neuroblastoma heterogeneity within the tumour tissue and across the course of disease in individual patients, we performed genomic and transcriptomic analysis of multiple spatially and temporally distinct samples per patient. We performed whole-exome sequencing (WES, $N = 51$ biopsies), targeted sequencing ($N = 140$ biopsies) and RNA sequencing ($N = 48$ biopsies) from 10 patients at the age of 0.5–14 years (3 girls and 7 boys) treated for neuroblastoma at the Charité Berlin, according to the German *NB2004* trial and *NB Registry 2016* protocols (Fig. 1 and Supplementary Fig. 1). Seven of the ten patients had a high-risk, 1 child an intermediate-risk and 2 patients a low-risk disease according to INRG/INSS classification. Between 2 and 10 biopsies per patient were analysed by WES (Fig. 1). For 3 patients, samples from 2 different time points were included. Samples were biopsied from primary tumours at diagnosis ($N = 17$), after 4–6 cycles of induction chemotherapy at tumour resection ($N = 26$) or at diagnosis of relapse ($N = 8$). Primary ($N = 45$) and metastatic tumour sites ($N = 6$) were biopsied (Fig. 1). To increase the sensitivity for SNV and SCNA detection, tumour regions with a high tumour purity (>60% for WES and RNA sequencing) were macrodissected from sequential sections. Matched normal blood controls were collected once from each patient at diagnosis. SNVs and SCNAs were called from WES data. To assess the clonality of SNVs detected with WES at higher resolution, we performed ultra-deep targeted sequencing (mean coverage 2500×). To this end, following WES analysis of 51 samples, we included further 89 samples analysing a total of 140 samples by ultra-deep targeted sequencing of 1479 WES-detected SNVs.

Consistent with previous reports[3,4,21,26], the total number of somatic, non-silent SNVs varied between 0 and 60 (median: 15) per sample (Fig. 1c). As expected, low-risk neuroblastomas had significantly fewer non-silent mutations than high-risk neuroblastomas ($p = 9 \times 10^{-5}$, two-sided Wilcoxon test)[3,10,21]. For each patient, SNVs were categorised as "clonal" if detected in all samples from that patient or "subclonal" if detected in a subset of samples from that patient or "subclonal/specific" if exclusively detected in one sample (Fig. 1c). The proportion of clonal SNVs varied from 0 to 87% among samples (average over patients: 37%), while on average 22% of mutations in each sample were categorised as subclonal/specific (Fig. 1c), pointing towards significant intratumour heterogeneity. SNVs were considered to be clonal if present with a variant allele frequency (VAF) greater than 10% in WES or targeted sequencing data in all samples. This threshold was chosen to avoid overestimation of heterogeneity. All other detected SNVs were considered subclonal. Roughly 45% of SNVs were clonal, irrespective of whether the SNVs were silent, non-silent or were detected within cancer-related genes (Supplementary Fig. 1b and Supplementary Data 1). For the large majority of SNVs (87.9%), assignment as clonal or subclonal were consistent between WES and targeted sequencing datasets (Supplementary Fig. 1c). As expected, the overall proportion of SCNAs markedly differed among patients. In line with previous reports, *MYCN* amplification, a hallmark of high-risk neuroblastoma, showed little variation between spatially distinct samples, but pronounced differences in patient CB1003 between time points (Fig. 1c). Our data demonstrate extensive spatial and temporal genetic heterogeneity across distinct neuroblastoma biopsies from the same patient in our cohort.

### Spatial and temporal heterogeneity of single-nucleotide variants affects therapeutically actionable genes in neuroblastoma.

To determine biologically and potentially clinically relevant

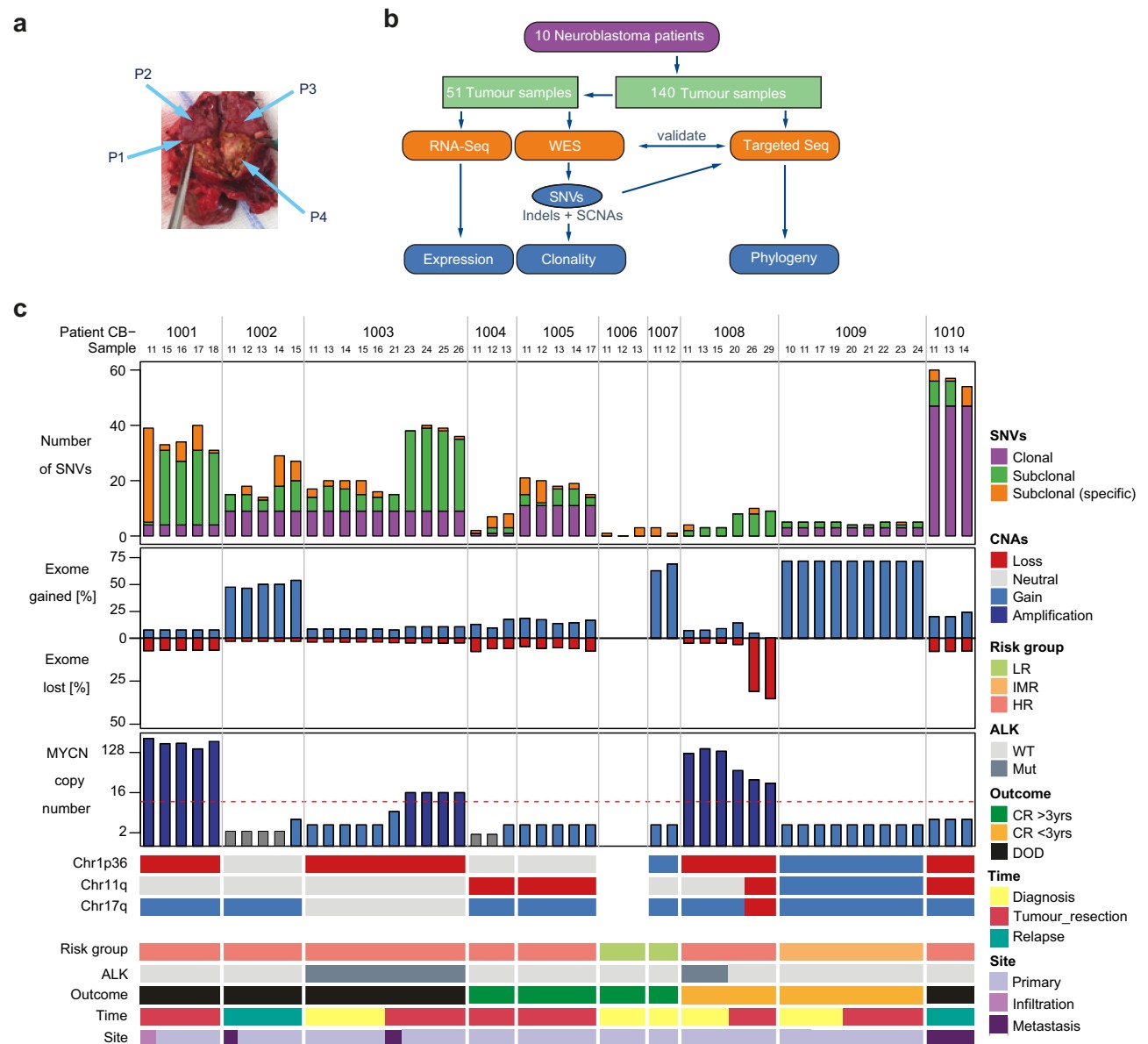

**Fig. 1 Study design and overview of neuroblastoma intratumour heterogeneity determined in 10 patients from multi-region WES and targeted re-sequencing. a** Representative picture of a neuroblastoma (arrows indicate spatially separated samples for analysis). **b** Study design and sample workflow. **c** Oncoplot outlining patient/sample characteristics and WES results from 51 neuroblastoma samples collected from 10 patients (CB1001 – CB1010). Columns correspond to individual samples. Gained regions are indicated in blue, lost regions are marked in red. Risk stratification determined patient risk at diagnosis as high (HR), intermediate (IMR) and low (LR) according to current practice. Patient outcomes are indicated, including death of disease (DOD) and complete remission (CR), with CR being >3 years for this cohort. SNV, non-synonymous single-nucleotide variants; SCNA, somatic copy-number alteration, clonal, present in all samples; subclonal, present in >1 sample from 1 single patient; subclonal (specific), present in a single biopsy.

genetic intratumour heterogeneity, we assessed non-silent SNVs in known cancer-related genes (Fig. 2). Whereas between two and six non-silent SNVs were detected in 801 selected known cancer-related genes (Supplementary Data 1 and Supplementary Table 2; defined by the COSMIC database[27] and other studies[8,10,16,28]) in high-risk tumours, very few or no non-silent SNVs in the selected cancer-related genes were detected in neuroblastomas from other risk groups (Fig. 2a). Most SNVs were detected in all samples (Fig. 2a) from a given tumour, indicating the clonal presence and early evolutionary emergence. Known pathogenic mutations in *HRAS* were among these clonal mutations, and a clonal *ALK* mutation was detected in patient CB1003. However, some detected mutations appeared spatially and/or temporally heterogeneous (e.g. *ALK* R1275Q in patients CB1002 and CB1008, as

well as *FGFR1* in patient CB1002, Fig. 2a). The persistence of clonal mutations in *ATM*, *PTK2*, *GLI3* and *CLTCL1* in all tumour locations analysed in samples from diagnosis and at relapse in patient CB1002 (Fig. 2b) could indicate the relevance of these SNVs for tumour progression. In contrast, high overall numbers of *de novo* SNVs and mutations in the druggable genes, *ALK*, *BRAF* and *FGFR1*, were only detected in a subset of relapse samples from patient CB1002 compared to the primary tumour at diagnosis (in line with previous studies reporting an increase in mutational burden and *de novo* MAPK pathway mutations in relapsed neuroblastoma[26,29]). The high degree of spatial hetero-geneity observed for SNVs in druggable target genes is exempli-fied by the *ALK* R1275Q mutation only being detected in 2 of 7 locations in the relapse tumour, the *BRAF* V600E mutation in

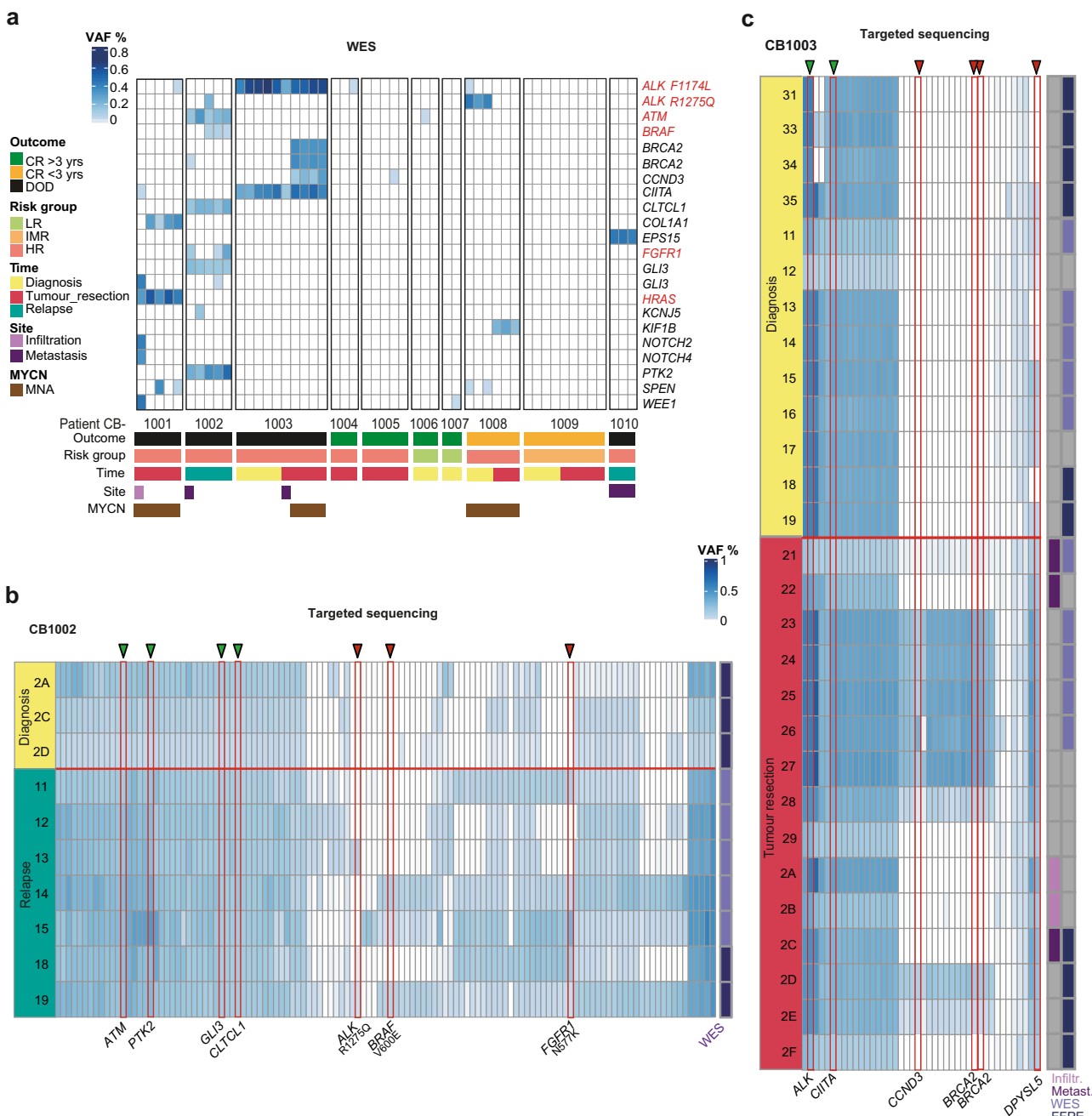

**Fig. 2 ITH of SNVs in cancer-related genes analysed by WES and ultra-deep targeted sequencing. a** Heatmap showing mutations in known cancer-relevant genes, based on WES of 51 tumour samples from 10 patients (CB1001 to CB1010). The frequency of mutated compared to wild-type alleles (VAF) is depicted in a blue colour code. Cancer-related genes connected to neuroblastoma genetics are marked in red. Single-nucleotide variants (SNVs) detected at separate positions in the gene are listed twice. Each column corresponds to an individual sample with samples from the same patient grouped next to each other. Clinical characteristics of patients and samples are annotated below. For 3 patients (CB1003, CB1008, CB1009), samples from different time points (Time) were analysed for the primary tumour site, distant metastasis or metastatic infiltration (Site). DOD: death of disease, CR: complete remission, HR: high risk, IR: intermediate risk, LR: low risk, MYCN: *MYCN* amplification. VAF of somatic SNVs based on targeted sequencing data from two exemplarily shown patients, CB1002 (**b**) and CB1003 (**c**). Only SNVs detected in one or more samples are displayed as one column per SNV. Mutations in cancer-related genes are explicitly named, and were indicated as clonal (VAF > 10% in all samples from a tumour, green triangles) in comparison to subclonal oncogenic mutations (red triangles). Row numbers identify samples at distinct time of biopsy collection within the disease course. Columns adjacent to the right of heatmaps indicate further characteristics. Samples denoted with WES were already included in the exome sequencing data. Samples denoted with FFPE were formalin-fixed and paraffin-embedded before analysis was performed, all other samples were fresh-frozen biomaterial.

only 3 of 7 locations and the *FGFR1* N557K mutation in only 5 of 7 locations. This detailed picture of spatial heterogeneity in actionable target genes in patient CB1002 points to a potential reliability problem for the current routine practice of target identification by single biopsy- and bulk sequencing-based

molecular tumour profiling as this profile may not accurately describe the disease in the patient. Spatial heterogeneity with specific mutations in the cancer-relevant genes, *FGFR1*, *ALK* and *ATM*, among others, was also detected in other patients in our cohort supporting that the spatial heterogeneity in druggable

targets clearly presented in the case of patient CB1002 is likely not a rare exception, creating the need to re-evaluate targeted therapy decision-making.

In another example, the actionable *ALK* R1275Q mutation was observed at high variant allele frequency at diagnosis, but not detectable by WES after chemotherapy in patient CB1008 (Fig. 2a). Targeted ultra-deep sequencing, however, was able to recover this mutation in the sample after therapy, albeit at very low allele frequency (Supplementary Fig. 2f). This was further validated by a droplet digital PCR (ddPCR) assay confirming high VAF of the ALK R1275Q at diagnosis but loss of the mutation in 8 out of 10 samples from the resection (Supplementary Fig. 3). A mutation in the *KIF1B* tumour suppressor was detected (WES) only after chemotherapy in patient CB1008, which was confirmed by targeted ultra-deep sequencing, suggesting that this mutation occurred *de novo* during therapy (Fig. 2a and Supplementary Fig. 2f). Finally, clonal expansion of a mutation after therapy was also observed in a subset of cases, such as the *DPYSL5*[30] mutation in patient CB1003 (Fig. 2c). We conclude that actionable and pathogenic SNVs are spatially and temporally heterogeneous in neuroblastoma, revealing clonal evolution prior to and under selective treatment pressure with potential implications for targeted therapy decisions.

**High degrees of spatial copy-number heterogeneity and ongoing chromosomal instability detected in a subset of neuroblastomas.** To investigate the degree of ongoing chromosomal instability in neuroblastoma, allele-specific SCNAs were called from WES data for all 51 samples from the 10 patients. Quality control revealed a low signal-to-noise ratio in patient CB1006, and this patient was excluded from further analyses, yielding a final set of 48 samples from 9 patients. Allele-specific SCNA profiles underwent reference phasing, phasing-based correction of copy-number states and detection for structural parallel evolution using *refphase* as previously described[23]. The phased and corrected allele-specific SCNA profiles included information on subclonal SCNAs derived from opposite alleles as mirrored subclonal allelic imbalances (MSAI) and were subjected to tree reconstruction using MEDICC2[31,32], including reconstruction of the most recent common ancestor to detect early events prior to clonal diversification.

To allow for consistent comparison of SCNA events across patients, SCNAs were aggregated on a per-cytoband level and stratified into clonal (present in all samples from that patient) and subclonal (present in some samples from that patient) events. Strong heterogeneity was detected on the level of SCNAs across the cohort. Across the cohort, between 13 and 92% (median 34%, mean 42%) of the genome of each patient's cancer was affected by at least one SCNA event in at least one sample (Fig. 3a and Supplementary Fig. 4). We detected subclonal SCNA events in every patient, with a median of 57 segments affected by subclonal SCNA events compared to 132 clonal events. The majority of SCNAs were clonal for most patients (CB1001, CB1002, CB1007, CB1009 and CB1010), indicating early structural evolution followed by relative stasis. Contrary to this, a similar number of clonal and subclonal SCNAs were detected in patients CB1003, CB1004 and CB1005, demonstrating ongoing clonal evolution and chromosomal instability (Table 1). Patient CB1008 showed a disproportionally large number of subclonal, compared to clonal events, indicative of strong clonal diversification and ongoing chromosomal instability between samples obtained at diagnosis and at tumour resection after first cycles of induction therapy. This patient showed repeated *copy-number neutral loss of heterozygosity* (LOH) events in large areas of the genome (chromosomes 7–9, 12, 14–16, 17 and 19) only in the tumour

resection samples, while sharing the clinically relevant SCNA events, LOH of 1p and *MYCN* amplification, among diagnostic and resection samples. Across the cohort, aggregated SCNA profiles show clear non-uniformity along the genome (Fig. 3) with hotspots in the clinically relevant regions, 1p36 (clonal LOH or clonal gain events in 8 of 9 patients), 11q (clonal LOH or gains in 5 of 9 patients) and 17q (clonal gain events in 8 of 9 patients). Interestingly, even though *MYCN* amplifications were detected in only 3 of 9 patients, we observed clonal gains of 2p24 in 6 additional patients, making the *MYCN* locus the most frequently gained region in our cohort even in low- and intermediate-risk patients. In line with previous reports[33], subclonal or clonal gains of chromosome 7q were detected in 8 of 9 patients. We also detected repeated MSAI events in 3 of 9 patients (Fig. 4) indicative of parallel evolution, but a low sample size prevented assessment of statistical significance.

The clinical importance of SCNAs in neuroblastoma strongly depends on the size of the events. Segmental SCNAs generally confer a poor prognosis whereas whole-chromosome gains and losses are considered neutral towards patient outcome[34–36]. To explore this, we stratified chromosomes into unchanged (diploid), whole-chromosome altered and segmentally altered chromosomes, and separately tallied their fractions in each category for each patient (Fig. 3b). The majority of chromosomes were affected by segmental duplications in most patients. In two patients (CB1007 and CB1009), however, most chromosomes were instead affected by whole-chromosome gains and losses (Fig. 3b). Patients CB1007 and CB1009 were the only low- and intermediate-risk patients in our cohort for whom SCNA data was available, confirming prior observations about the clinical relevance of segmental versus whole-chromosome SCNAs in neuroblastoma[37,38]. In summary, SCNAs are highly spatially and temporally heterogeneous in neuroblastomas, and in some cases are continuously acquired, indicating ongoing chromosomal instability.

**Inference of evolutionary trajectories allows relative timing of the emergence of the metastases in neuroblastoma evolution.** We next investigated each patient individually for evidence of early clonal diversification by inferring clonal hierarchies from variant allele frequency-based SNV clustering. We observed both linear (CB1003) and branched (CB1002) hierarchies in high-risk patients (Fig. 4a), supporting variable subclonal diversification during high-risk disease. Clonal evolution based on successive mutation acquisition in different spatial and temporal samples from each patient led to between three and eleven genetically distinct subclonal branches and lineages (clone numbers in circles, Fig. 4a). In some cases, samples from a patient (numbers at the lineage ends, Fig. 4) are assigned to different subclonal lineages indicating that the biopsy sample harboured multiple subclones (i.e. samples 14, 15 from CB1002). This clonal intermixing of subpopulations within one sample illustrated the extensive spatial and temporal genetic heterogeneity in our cohort (Fig. 4a). In 7 out of 8 cases for which both SNV and SCNA data were available, tumour phylogenies based on SCNA profiles agreed with the hierarchies inferred from SNVs (Fig. 4 and Supplementary Fig. 4). In CB1005 the SNV and SCNA topologies did not agree. Here the SCNA phylogeny groups the dominant clones of samples 12 and 17 as well as those of 11, 13 and 14 into a monophyletic clade (Supplementary Fig. 4). Ancestral reconstruction using MEDICC2 provided additional evidence in support of this topology, including a gain on chromosome 6q shared between 12 and 17 and an additional gain on 7q shared between 11, 13 and 14. The analysis also revealed multiple parallel events, including MSAI on chromosomes 1q and 18p and 18q. We next compared samples from

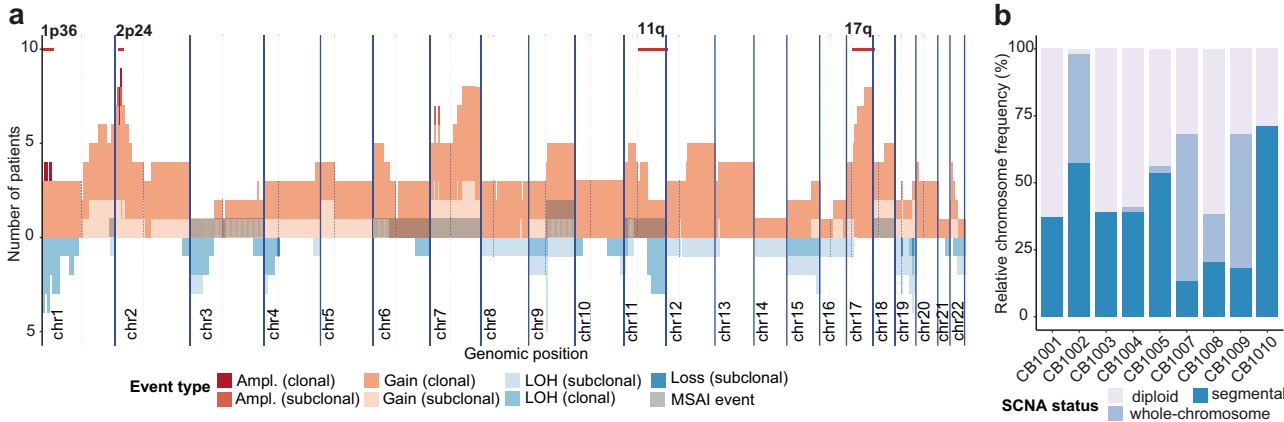

**Fig. 3 Intratumour heterogeneity of somatic copy-number alterations in the neuroblastoma cohort. a** Overview of Somatic copy-number alterations (SCNA) events per-cytoband aggregated across 9 patients. Amplifications and gains are visible in the upper half, loss of heterozygosity (LOH) and deep losses in the lower half of the plot. Clinically relevant chromosome regions 1p36, 2p24, 11q and 17q are highlighted in red at the top. Mirrored subclonal allelic imbalances (MSAI) are indicated by grey shading. *MYCN* locus: 2p24 **b** Fraction of chromosomes affected by segmental or whole-chromosome gains and losses are depicted for the low-and intermediate-risk patients (CB1007, CB1009) and the high-risk patients (CB1001, CB1002, CB1003, CB1004, CB1005, CB1010).

**Table 1 Clonality of somatic copy-number alterations identified in distinct neuroblastoma samples during the course of disease in 9 patients.**

| Patient | Risk group | # Clonal segments (% genome) | # Subclonal segments (% genome) | Biopsied samples | # Samples | MNA |
|---------|-----------|------------------------------|----------------------------------|------------------|-----------|-----|
| CB1001 | HR | 100 (12%) | 3 (0.4%) | Tumour resection | 5 | Yes |
| CB1002 | HR | 644 (79%) | 105 (13%) | Diagnosis of relapse | 5 | No |
| CB1003 | HR | 54 (7%) | 57 (7%) | Neuroblastoma diagnosis + tumour resection | 10 | Yes |
| CB1004 | HR | 104 (13%) | 100 (12%) | Tumour resection | 3 | No |
| CB1005 | HR | 132 (16%) | 98 (12%) | Tumour resection | 5 | No |
| CB1007 | LR | 487 (60%) | 17 (2%) | Neuroblastoma diagnosis | 2 | No |
| CB1008 | HR | 24 (3%) | 353 (44%) | Neuroblastoma diagnosis + tumour resection | 6 | Yes |
| CB1009 | IMR | 561 (69%) | 1 (>0%) | Neuroblastoma diagnosis + tumour resection | 9 | No |
| CB1010 | HR | 247 (30%) | 33 (4%) | Diagnosis of relapse | 3 | No |

*HR* high-risk, *IMR* intermediate risk, *LR* low risk, *MNA* MYCN amplification.

primary and metastatic tumour sites. CB1001 is a patient with high-risk disease that includes a clonal *MYCN* amplification, and for whom 5 SCNA profiles were available comprising 4 regions from the primary tumour and 1 from a metastatic renal infiltration. This metastatic infiltration probably occurred not by hematogenous dissemination but by aggressive outgrowth of the primary tumour. Limited heterogeneity at both SCNA and SNV levels occurred in the CB1001 primary tumour sites, and most alterations were clonal (Fig. 4b). Both SCNA and SNV data indicated branching of the metastatic clone prior to the emergence of the clones in the primary (Fig. 4b, c and Supplementary Fig. 2a). This "early" branching is clearly evidenced by parallel evolution between the primary and metastatic clades: chromosome 9 harboured an MSAI event in which a single additional copy of 9q was gained in both primary and metastatic samples, but the copies originated from different parental chromosomes (haplotypes) in the metastatic infiltration and primary tumour (Fig. 4e). In addition, all biopsies from the primary tumour also harboured an ~2.5 mb focal LOH deletion event on chromosome 6q, which was absent in the metastatic infiltration (Fig. 4d). Since lost genetic material cannot be regained, this clearly indicates the branching of the metastatic clone before the diversification of the primary

samples. The tree topology additionally showed high robustness as measured by jackknife resampling of the underlying SCNA data (Fig. 4c). A similar pattern emerged in patient CB1003, from whom 10 samples were available (5 biopsies at diagnosis, 4 from the resected tumour and 1 synchronously resected lymph node metastasis). SCNA phylogeny clearly separated the two time points (Fig. 4g) and showed a dynamic acquisition of additional *MYCN* copies during disease progression (Fig. 4h and Supplementary Fig. 5). Multiple single-copy gain events occurred on 2p24 in the primary tumour at diagnosis that developed into a *MYCN* amplification (16 copies) at the time of tumour resection (Fig. 4f, g, h and Supplementary Fig. 5). An intermediate level of *MYCN* gain, estimated at 6 copies, was detected in the metastasis (Fig. 4h), indicating the initial 2p24 gain occurred early in tumour evolution, followed by additional 2p24 gains in the primary tumour and the metastasis, likely due to increased accumulation of *MYCN* in circular extrachromosomal DNA elements, as recently described[39–41]. Interestingly, chromosome 5q contained a single-copy gain event clonal in all samples but the metastasis. Two hypotheses are thus supported by the data: (i) emergence of the metastasis before diversification of all primary samples followed by parallel gains on 2p24, or (ii) branching of the metastasis after

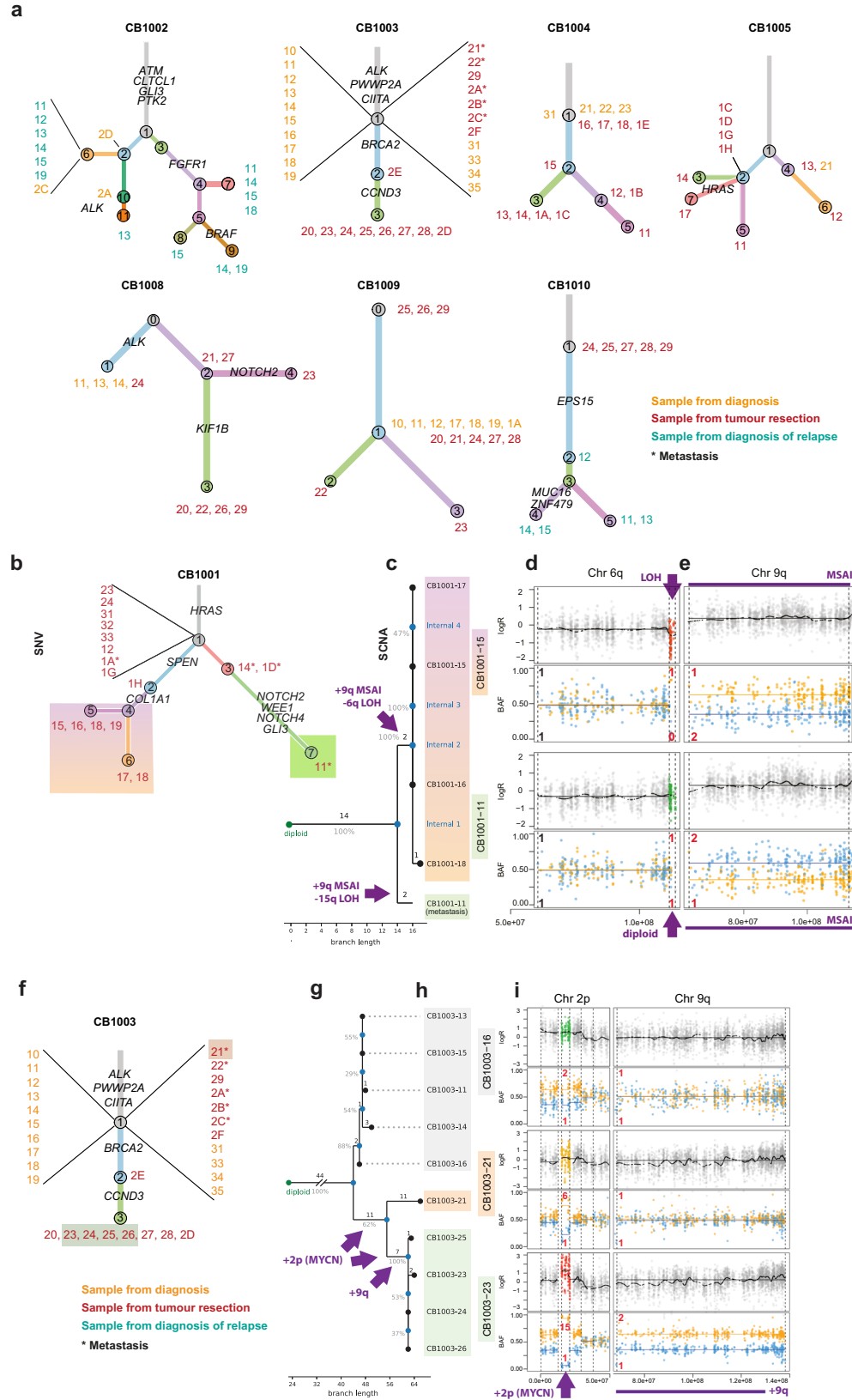

the emergence of the clone in the biopsy samples but before diversification of the resection samples (Fig. 4) implicating a secondary loss of 5q in the metastatic branch. Tree reconstruction with MEDICC2 inferred option (ii) as the more parsimonious (Fig. 4) and jackknife robustness analysis of the tree indicates a

62% support for the metastatic split between biopsy and resection samples. In any case, all samples obtained from the resected tumour showed clear allelic imbalance and a single-copy gain of chromosome arm 9q, which was absent in both the diagnostic biopsies and the metastasis resected synchroneously with the

**Fig. 4 Primary and metastatic tumour evolution inferred by genetic analysis of clonality. a** Single-nucleotide variant- (SNV)-based phylogenetic trees depict variable clonal diversification. Branch lengths correspond to the total numbers of SNVs. Branch colours represent different subclones. Numbers at the end of each branch indicate samples containing all clones (numbers in circles) of the branch and its ancestor clones. Samples were collected at diagnosis (orange), at tumour resection (red) and at diagnosis of relapse (turquoise). Metastatic samples are indicated by asterisks. A deeper data showcase is provided for patient CB1001, showing SNV- (**b**) and somatic copy-number alteration- (SCNA)-based trees **(c). d** SCNA plots from sample 15 (rainbow) showing a loss of heterozygosity (LOH) event on chromosome 6q that is absent in the metastatic sample 11 (green). **e** SCNA plots from the metastatic sample 11 (green shaded) and primary tumour sample 15 (rainbow) showing an mirrored subclonal allelic imbalances (MSAI) event on chromosome 9. SNV tree **(f)** compared to SCNA phylogeny **(g)** in CB1003. Time point or spatial position is indicated by shading, none (metastasis), grey (biopsy at diagnosis) and green (tumour resection). **h** Dynamic acquisition of additional copies of the *MYCN* locus on chromosome 2p. **i** Monoallelic gain on chromosome 9q in the tumour resection samples. (green) but not in the metastatic clone.

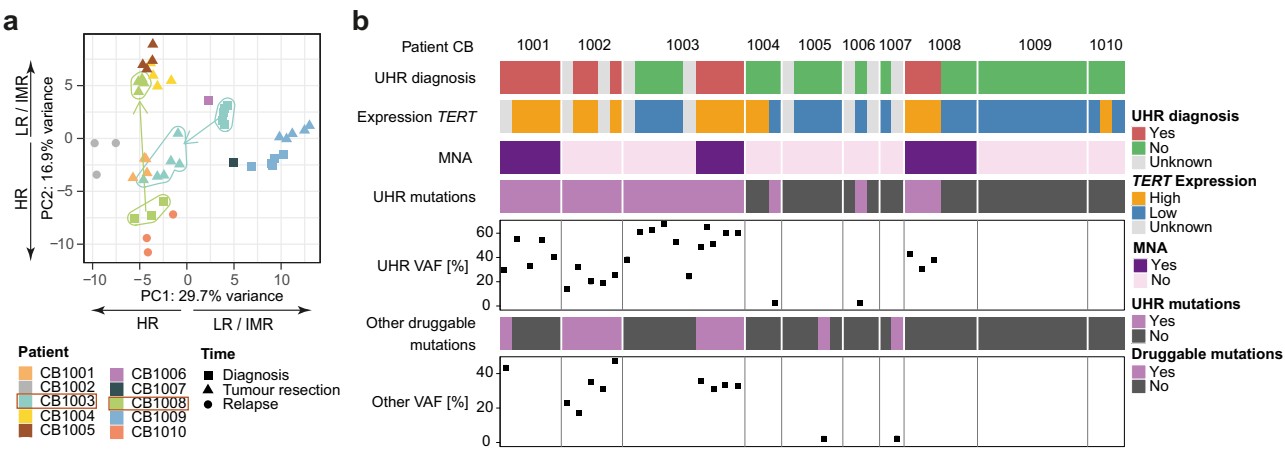

**Fig. 5 Impact of temporal and spatial intratumour heterogeneity on gene expression, ultra-high-risk stratification and putative therapy options. a** RNA sequencing was used to analyse gene expression in 48 spatially and temporally distinct samples from a cohort of 10 patients. Principal component analysis (PCA) was conducted for risk-associated gene expression in the 48 samples. Legend describes patient number and sampling time. Arrows within the plot indicate samples from patients CB1003 and CB1008 collected at different time points during the disease. Arrows adjacent to the axis annotate high-risk and low-risk/intermediate-risk neuroblastomas. **b** Molecular classification of ultra-high risk (UHR) for the patient (termed UHR diagnosis here) was performed for each tumour sample separately using *TERT* mRNA expression, *MYCN* copy number and mutations (WES) in the TP53 and RAS/MAPK pathways. Variant allele frequency (VAF) is given for all mutations as % of reads that support the mutation. Mutations in the TP53/RAS/MAPK pathways with a VAF > 5% were defined as UHR mutations. Other druggable mutations based on the INFORM criteria[8] are shown for each sample. *TERT* expression was defined as "high" according to Ackermann et al.[3].

primary tumour (Fig. 4i), confirming additional events and ongoing evolution after biopsy and metastatic dissemination. Thus, SCNA analysis indicates high genomic divergence between primary tumour and metastases in neuroblastoma suggesting that metastases spread early in disease evolution from the primary tumour site, although additional studies are required to further substantiate this.

**Gene expression is temporally but not spatially heterogeneous in neuroblastomas.** To detect phenotypic heterogeneity in spatially and temporally distinct neuroblastoma samples, high-coverage paired-end messenger RNA sequencing was performed on 48 of 51 samples and gene expression was quantified (Figs. 1b, 5a and Supplementary Fig. 6). Genes known to be specifically expressed in neuroblastoma, such as *HAND2, PHOX2B*, and *DBH*, were homogeneously expressed across all samples, indicating that the proportion of mRNA derived from neuroblastoma cells was equally distributed between each sample. RNA expression analysis confirmed the impact of the most frequent copy-number changes over time in the cohort, i.e. the *MYCN* amplification, and revealed a clear correlation between *MYCN* copies and gene expression of *MYCN* and of its target gene *TERT*. Other SCNAs also clearly correlated with changes in gene expression, such as the gain in chr 7, which was detected in 8 out of 9 patients and which led to an increase of expression of the

*IL6* gene located on chr. 7p15.3 (Supplementary Fig. 6a). To further address the consequences of temporal ITH of SCNAs, we performed Gene-Set Enrichment Analysis (GSEA) of known downstream targets of *MYCN*[42] (Supplementary Data 2) for patients CB1003 and CB1008. In CB1003, we found *MYCN*-regulated genes enriched towards the tumour resection in line with the acquisition of the *MYCN* amplification at that time point. In CB1008, *MYCN*-regulated genes were significantly enriched towards diagnosis, in agreement with the decrease of *MYCN* copy number during tumour progression (Supplementary Fig. 6b). Based on bulk RNA-seq data, temporal expression changes between diagnostic versus tumour resection did not include a significant shift in differentiation state-related signatures i.e. mesenchymal versus adrenergic signatures[43] (Wilcoxon test *p*-values > 0.05) (Supplementary Fig. 6c). A principal component analysis (PCA) utilising expression levels of the most variable genes confirmed that interpatient differences dominated over intratumour heterogeneity in gene expression (Supplementary Fig. 6g and Supplementary Data 4). A different picture emerged when PCA was restricted to genes previously used for expression-based classification between high- and low-risk neuroblastomas[44] (Supplementary Data 3). Low- and high-risk tumours in our cohort were clearly separated, as expected, but two sets of samples collected at different time points in three patients were also clearly separated (Fig. 5a). In patient CB1003, samples from the diagnostic biopsy clustered with low-risk

tumours, while the samples from the resected tumour clustered with high-risk samples, suggesting therapy-induced changes in the expression of risk-associated genes (Fig. 5a). Our data demonstrate that gene expression programmes in neuroblastomas clearly follow temporal heterogeneity in SCNA and can change significantly under therapy.

**Temporal and spatial genetic neuroblastoma heterogeneity in a patient affects clinical assessment of ultra-high risk and second-line therapy options.** We and others have recently described molecular determinants of clinical UHR stratification[3]. This UHR stratification is based on (i) the presence of telomere maintenance via *TERT* rearrangement, *MYCN*-amplification, high *TERT* expression, or ALT and (ii) mutations in TP53 and RAS/MAPK pathways. We observed high spatial intratumour genetic heterogeneity and subclonal evolution in our patient cohort that affected these UHR determinants (Figs. 1–4) and hypothesise that clinical risk stratification may be affected by this genetic heterogeneity. To test this, we performed molecular-based risk stratification on each of the neuroblastoma biopsies separately using the information on *TERT* mRNA expression, *MYCN* copy number and TP53/RAS/MAPK pathway mutations from WES data. Based on these molecular risk factors, almost one third (29%) of biopsies (15 of 51) were classified as UHR (Fig. 5b). In line with our hypothesis, not all biopsies from the same patients were classified homogenously. Whereas biopsies collected at diagnosis in patient CB1003 were not classified as UHR, all biopsies from the primary tumour resected after induction therapy were classified as UHR (Fig. 5b). This suggests the clonal selection of UHR subclones during therapy. In contrast, diagnostic biopsies from patient CB1008 (currently in complete remission for >3 years) were classified as UHR, while biopsies after chemotherapy at tumour resection were not classified UHR. Most prominently, this occurred due to longitudinal changes in *MYCN* copy number (CB1003) or *ALK* mutation as well as *TERT* gene expression (CB1008). In total, 2 out of 10 patients showed a heterogeneous UHR profile. This is important because the neuroblastoma community is currently discussing to base alternative experimental therapy decisions on the presence of druggable mutations and UHR classification at diagnosis in the future, but our data clearly show that a subset of UHR patients still benefits from standard high-risk treatment. We conclude that clonal evolution in neuroblastoma can influence the molecular profile within the tumour, across disease sites and/or during the course of disease in potentially clinically relevant ways. Our findings support the consideration of strategic and methodological changes for target identification and molecular risk classification as well as subsequent treatment decisions in individual patients with neuroblastoma.

## Discussion

Intratumour heterogeneity is considered a major cause of cancer progression and treatment resistance[24]. Multi-region sequencing studies have characterised the extent of intratumour heterogeneity in a variety of cancer entities, and linked the evolutionary subtypes with patient outcome[24,35,45,46]. Our present multi-region sequencing study of 10 patients with neuroblastoma shows extensive spatial and temporal intratumour heterogeneity that also affected genes encoding actionable targets, such as *ALK* and *FGFR1*, thus, harbouring potential implications for therapeutic target selection. Analysis of phased SCNAs revealed early emergence of the metastatic lineage and ongoing chromosomal instability, including the appearance of a *MYCN* amplification later in the course of disease. This process of clonal evolution led

to clinically relevant changes in molecular risk stratification in some patients.

Our in-depth analysis of SNVs and SCNAs assessed a significantly lower number of SNVs, and practically no mutational hits in cancer-related genes in low- and intermediate-risk neuroblastomas. Overall, we detected a low number of non-silent SNVs (range 0–60, median 15), which is in accordance with previous reports[3,4,26]. Less than half of the detected SNVs were clonal, based on our criteria for clonal appearance (present in all samples from a patient with VAF > 10%). To gain insight into the biological and clinical relevance of intratumour heterogeneity, we focused our analysis on non-silent SNVs in cancer-related genes. Non-silent SNVs (2–6 per tumour) in cancer-related genes (*HRAS*, *ALK* and *ATM*) occurred exclusively in high-risk neuroblastomas and were clonal in a subset of patients but also spatially or temporally heterogeneous in others.

Efficacy of a targeted therapy that might be included into second-line treatment for a patient with neuroblastoma depends on precise and reliable diagnostics. We detected spatially heterogeneous SNVs in genes encoding actionable targets (*FGFR1* N577K, *BRAF* V600E and *ALK* R1275Q) or temporally heterogeneous SNVs in *CCND3* or *NOTCH2*. Spatially heterogeneous mutations suggest that current diagnostic biopsies may miss or overestimate therapeutically actionable alterations. More specifically, application of current target prioritisation algorithms might have obtained different results when analysing different tumour locations, such as relapse tumour samples from our patient CB1002, which harboured different private druggable mutations. Most importantly, even the presence of activating *ALK* mutations, the most frequent actionable alteration in neuroblastoma[9,35,47] for which targeted therapies are currently being clinically tested[48–53], was spatially and temporally heterogeneous. Such heterogeneity could impact the clinical interpretation of diagnostic biopsies and may obscure reasons for nonresponse to *ALK* inhibitor therapy in patients in whom the majority of the tumour cells do not contain activating *ALK* mutations[54]. Our finding that druggable mutations are spatially and temporally heterogeneous demonstrate that clonal neuroblastoma evolution has potential implications for targeted therapy decisions. We postulate that for a subset of patients, bulk sequencing at a single tumour location might sometimes lead to an imprecise molecular diagnosis, potentially leading to the wrong choice for targeted therapy. In the future, the technological advances of single-cell technologies and liquid biopsies might contribute to solve this problem, as repeated multi-region sequencing of tumour tissue is not feasible in the clinical routine.

Allele-specific SCNA profiling and multi-region phasing based on WES data revealed substantial heterogeneity in neuroblastoma chromosomal instability in patients in our cohort. Subclonal SCNAs were observed in every patient and 3 out of 9 patients showed evidence of parallel evolution through detection of MSAI events. Most SCNAs were numerical chromosomal aberrations in the two patients with low- and intermediate-risk disease, correlating with previous reports[37,38]. In most patients (5 of 9), the majority of the SCNAs were clonal, demonstrating the early appearance of SCNAs in the course of disease. Interestingly, in one patient with mostly clonal SCNAs, extensive intratumour SNV heterogeneity was detected, suggesting that in this patient SCNAs were the tumour-driving force followed by subclonal mutational diversification. In the remaining patients, variable degrees of ongoing clonal evolution and chromosomal instability occurred. Strong temporal heterogeneity was observed in one patient (CB1008) with many subclonal SCNAs and ongoing chromosomal instability between samples collected at initial diagnosis or after therapy, but stable detection of a *MYCN* amplification and loss in 1p36 both at neuroblastoma diagnosis

and following therapy. The heterogeneous profile of cancer-related gene mutations in the same patient showed the near elimination of an initial *ALK* R1275Q mutated clone after therapy, further corroborating the extensive temporal heterogeneity. To better understand neuroblastoma evolutionary diversification in each patient, clonal hierarchies were reconstructed from SNVs in biopsies from different time points and spatial distributions of the disease. We observed both linear and branched evolutionary trees, creating three to eleven distinct lineages per patient, and indicating variable clonal diversification across the cohort. Previous studies[13,14] defined four evolutionary trajectories over different anatomic areas for childhood cancers, among them 23 neuroblastomas including two trajectories correlating with poor outcome. We frequently observed simultaneous subclonal SCNAs, clonal segmental SCNAs and clonal or subclonal cancer gene mutations in the same high-risk neuroblastoma, tracing a clonal sweep as was previously described[14]. Due to our small cohort size of 10 patients, correlation with outcome was not conclusive. In contrast to prior reports[14], our allele-specific SCNA analysis detected clinically relevant 1p and 11q losses as mostly clonal while in one of three cases with *MYCN* amplifications, the amplification was not clonally persistent but occurred later during the course of disease. Taken together, the high degree of both spatial and temporal neuroblastoma heterogeneity and the multiple evolutionary trajectories observed to date in individual patients with neuroblastoma indicate that many complex routes culminate in severe neuroblastoma courses and will require detailed and precise molecular analyses in further patients to completely understand causal or key points in the process and to base second-line treatment selection on reliable molecular data for every individual patient.

To uncover the evolutionary processes of metastasis, we analysed synchronously resected primary tumour samples and metastatic infiltrations. Both SCNA and SNV data indicated branching of the metastatic clone during disease progression and divergent evolutionary trajectories, with clear evidence of clonal aberrations within the metastasis that were absent in the primary tumour. This was verified by MSAI events on chromosome 9q, allele-specific LOH or *MYCN* amplifications that differed between the metastases and matched primary tumours. In contrast, Karlsson et al.[14] found that synchronous metastasis mirrored the primary tumour. Branching of the metastatic clone and substantial heterogeneity between a metastasis and primary tumour may explain why neuroblastoma relapses often occur at metastatic sites while primary tumours are successfully eradicated by surgery and first-line chemotherapy. This is also supported by the high number of *de novo* mutations in relapse cases in our cohort, which indicate a major genotypic shift during (metastatic) recurrence, in line with the previous reports[14,26,29]. Inferring evolutionary trajectories from SNV and SCNA data reveals variable degrees of clonal diversification and an early emergence of metastatic clones in neuroblastoma.

An UHR group has recently been described based on molecular determinants[3,26] for a new concept of clinical risk stratification of patients with neuroblastomas. Notably, UHR molecular determinants were not homogenous throughout biopsies from the same patient in our cohort. This was especially true for patients CB1003 and CB1008, whose samples showed extensive temporal SNV and SCNA heterogeneity. Distinguishing UHR from HR disease may have important clinical consequences, since UHR patients have poor outcomes and may benefit from an earlier switch to alternative treatment approaches[55]. Based on our observation of spatial and temporal genetic heterogeneity of UHR and HR determinants in individual neuroblastoma cases, we believe that precise risk stratification can only be obtained through molecular characterisation of neuroblastomas at diagnosis, resection and recurrence. Our in-depth analysis of neuroblastoma intratumour heterogeneity reveals that extensive genetic heterogeneity and subclonal diversification evolves under therapy and may have important implications for the clinical interpretation of molecular diagnostic results and for the selection of appropriate second-line treatment approaches.

## Methods

**Sample collection**. Tumour specimens were collected from 10 patients enrolled between 2014 and 2018 in the German Society for Pediatric Oncology and Hematology (GPOH) NB2004 trial or NB registry 2016, and treated at the Charité University Medicine Berlin (Germany) according to the trial/registry protocols. All patients and/or their guardians gave written consent for the use of biosamples and clinical data for research in accordance with the local ethics review board of the medical faculty, University of Cologne as the trial sponsor of the NB2004 and the NB registry 2016 (https://clinicaltrials.gov/NCT03042429). Samples were collected by open surgical biopsy either at diagnosis, at tumour resection after 4–6 cycles of chemotherapy (according to neuroblastoma therapy regimen) or at diagnosis of relapse. Fresh samples were immediately snap-frozen in liquid nitrogen and stored at −80 °C. Portions of tumour material were formalin-fixed and paraffin-embedded (FFPE) in parallel for diagnostics and preservation in the pathology unit. Two to 30 biopsies were taken from geographically separate areas of the tumour body with a minimal distance of 10 mm from each single tumour. From these biopsies, only tumour regions with high tumour cell content (>60% for WES and RNA sequencing, 10% for targeted re-sequencing) were included. A pathologist confirmed the diagnosis and assigned tumour regions for macrodissection with a high content of vital tumour cells on sequential hematoxylin and eosin-stained sections. Peripheral blood collected from each patient was used as a matched germline control for tumour samples. DNA was prepared using the Qiagen DNA Mini kit according to the manufacturer's instructions.

**Compiling a list of relevant cancer genes**. A list of genes potentially involved in neuroblastoma development was compiled by combining lists of potential cancer driver genes from several sources: the COSMIC database[27] (version 80), a recent review of the mutational landscape in neuroblastoma[10], pan-cancer studies of paediatric tumours[8,16] and a pan-cancer study of adult cancers[28]. The final list consisted of 801 genes.

**Whole-exome sequencing (WES)**. The SureSelect Human All Exon V6 kit (Agilent) was used to prepare libraries enriched with exonic sequences. The libraries were further prepared for sequencing using the Illumina TruSeq Exome Kit and sequenced on Illumina HiSeq 2500 and Illumina NextSeq sequencers.

**Raw data generation and quality control**. Read sequences and base quality scores were de-multiplexed and stored in Fastq format using the Illumina *bcl2fastq (2.16)* software. Adapter remnants and low-quality read ends were trimmed off using custom scripts. The quality of sequence reads was assessed using *FastQC (0.11.5)* software. All libraries were deemed of high quality and usable for follow-up analyses.

**Alignment and coverage evaluation**. Reads were aligned to the human genome (assembly GRCh38) using *bwa mem (0.7.17)* software[56]. Duplicate read alignments were marked using *samblaster (0.1.24)*, and resulting Bam files were stored for later analyses. Overlap with exon target regions was assessed using the *BedTools (2.27.0)* suite of tools and the target region file provided with the SureSelect Human All Exon V6 kit (Agilent). Mean coverage was computed from these results as the average number of aligned reads per base in the specified exonic target region as were the exact proportions of targeted bases covered by each number of aligned reads. Median coverage was 346× (range: 163×–636×). Aligned reads were further filtered using *sambamba (0.6.8)* software to remove the secondary and failed alignments and PCR-duplicate reads.

**Somatic variant calling**. We determined somatic single-nucleotide variants (SNVs) and short insertions/deletions (InDels) using EBCall (2) software[57]. Hereby, each tumour sample is first compared to the matching control sample, and SNVs and InDels supported by a sufficient fraction (≥10%) of aligned reads in the tumour sample and not exceeding a specific fraction of reads in the control sample (<5%) are selected as candidate variants. Based on the complete set of control samples from all patients, a probabilistic model for base mismatches due to technical reasons was constructed, and only candidate variants deemed highly unlikely ($p < 0.001$) under this model, and thus, more likely to have arisen from biology rather than technical artifacts are considered as somatic variants. A SNV was considered to be clonal if present in all tumour cells with variant allele frequency greater than 10%, based on the assumption that all somatic mutations were heterozygous and tumour cell content in the sample was ~60%. The threshold was adjusted accordingly for SNVs in regions with copy-number gains and losses.

**Annotation of somatic variants**. The effect that each putative somatic variant would have on genes/proteins was annotated using *SnpEff (4.3t)* software[58] and the *knownGene* gene annotation from the UCSC genome browser for the hg38 assembly. For each variant, genes potentially affected by the mutation including effects on the protein level were stored together with information including whether the variant had been previously detected and deposited in a variant database such as dbSNP. A smaller list of variants predicted to affect well-known cancer genes was generated that only included non-silent variants in genes contained in the list of cancer-relevant genes described above.

***ALK* mutation detection using droplet digital PCR (ddPCR)**. The *ALK* R1275Q (3824, G>A) hotspot mutation and corresponding wild-type *ALK* sequence was detected using the QX200 Droplet Digital PCR System (Bio-Rad Laboratories, Munich, Germany). The AluI restriction enzyme (5 U, New England Biolabs, Frankfurt/Main, Germany) was added to each ddPCR reaction to fragment the tumour DNA[59]. Reaction mixtures for duplexed TaqMan ddPCR used 2× ddPCR *Supermix for Probes, no dUTP* (Bio-Rad Laboratories) and primers/probes adapted from Combaret et al.[60], and have been previously described[61]. Primers and probes had the following sequences: $ALK^{1275}$-for, 5'-GTCCAGGCCCTGGAAGAG-3'; $ALK^{1275}$-rev, 5'-GGGGTGAGGCAGTCTTTACTC-3'; $ALK^{R1275Q}$ probe, FAM, 5'-FAM-TTCGGGGATGGCCCAAGACAT-BHQ1-3' and $ALK^{1275}$ probe, HEX, 5'-HEX-TTCGGGGATGGCCCGAGACAT-BHQ1-3'. Manufacturer recommendations for the QX200 Droplet Generator were followed to create droplets, and reaction endpoints were assessed in the QX200 ddPCR Droplet Reader. The fraction of mutant allele was identified using QuantaSoft Analysis software, version 1.7.4.0917. Appropriate non-template, positive and negative controls were included in each ddPCR assay run for software-based generation of specific thresholds. The model of Armbruster and Pry[62] was followed to calculate (using Bio-Rad look-up tables) the false-positive rate (FPR) and detection limit for point mutation analysis. FPR calculation for the point mutation has been previously described[61], and was based in principle on the minimally required mutant target molecule concentration (copies/μl) and number of false-positive droplets. When the number of droplets detecting the mutation and the mutant target molecule concentration (copies/μl) were both above the thresholds that were set, a sample was scored as positive. Detection limit was determined as previously described[61].

**Determining allele-specific somatic copy-number alterations**. WES read counts from all neuroblastoma and matched germline (blood) samples were aggregated at 20,180,418 common germline variants from the NCBI dbSNP database using *samtools mpileup (1.9)*. Variants were removed that (i) were in highly variable regions of the major histocompatibility complex (MHC, chromosome 6: 28,510,120–33,480,577), (ii) were in positions characterised as not uniquely mappable by the UCSC Genome Browser annotation tracks (hoffmannMappability, k100.Unique.Mappability.bb)[63] and (iii) with <20 reads across both neuroblastoma/germline samples. Average genome-wide GC content was calculated in 50 bp windows using *sequenza-utils gc_wiggle*[64]. Read counts and GC content were forwarded to *Sequenza* (v3.0.0)[64] for segmentation and calling of allele-specific SCNAs in each biosample. Segmentations were combined for samples from each patient using a joint consistent segmentation approach as previously described[23,25]. The jointly segmented allele-specific SCNAs then underwent multi-region reference phasing using *refphase*[25] to determine haplotype-specific SCNAs and detect MSAI events indicative of parallel evolution. All haplotype-specific SCNA profiles were manually reviewed and individual mis-segmentations and erroneous sample purity and ploidy estimates were adjusted accordingly. Haplotype-specific SCNA profiles were then used to infer tumour evolution and reconstruct ancestral genomes using MEDICC2[31,32].

**Targeted sequencing**. A set of 1479 SNVs were selected from exome sequencing data for targeted sequencing at higher read coverage and in additional samples. This set included all SNVs in cancer-related genes (from our constructed list, Supplementary Data 1) and SNVs present in all patients to ensure that each patient had a minimum number of specific SNVs represented in the targeted sequencing panel. A custom targeted kit (Agilent) was used to enrich DNA from the genomic positions of these 1479 sites. DNA was isolated from 150 samples from the 10 patients, subjected to the targeted DNA enrichment, and sequencing libraries were generated for Illumina HiSeq and NextSeq sequencing (150 nt single-end reads). Reads were aligned to the human genome (assembly GRCh38) using *bwa mem* software[56]. Variant allele frequencies for each of the 1479 SNVs in each sample were computed using the *Rsamtools* Bioconductor/R package.

**Reconstruction of clonality trees**. The *sciClone* software (version 1.1.)[65] was used to infer clonal composition and possible neuroblastoma phylogeny in each patient. This method uses a Bayesian mixture modelling approach to stratify variants into clusters based on their copy number and variant allele frequency (VAF) in each sample. Sample tumour purity is estimated from the VAF distribution in copy-number neutral regions and used to scale all VAFs. The software setting, binomial-mixture model was used to cluster VAFs, with an upper limit set to 20 potential clusters and the default settings for other parameters.

VAFs of exactly 0 were replaced by a very low number ($10^{-10}$) prior to running the algorithm, since the clustering algorithm did not converge in extreme cases in which too many VAF entries were exactly 0. Variant clusters from *sciClone* were assigned to clones, and parental relationships between clones were inferred as follows. Clusters with high cellular fraction (>90%) in all samples were merged into the founding clone to make up the branch of the tree. Other clusters were grouped according to their cellular fraction in subsets of samples and assigned as clones to the branches of the tree, with the requirement that the ancestor clone was present in the same and potentially more samples. Clusters only present in a single sample were merged into clones that made up the leaves of the tree. Clones with a cellular fraction of <5% in a sample were not assigned to the respective sample. Samples were annotated in the tree at the relevant clone furthest away from the tree stem, indicating that the sample contained the respective clone and all of its ancestor clones. Clones were denoted as numbers, branch lengths corresponded to the total numbers of SNVs in each clone and cancer-relevant genes contained in each clone were annotated at the branches. For patients CB1005 and CB1006, the total numbers of SNVs with median VAFs >0 were too low to infer meaningful clonality trees.

**RNA sequencing**. Sufficiently high-quality RNA was available for 48 of 51 samples used for exome sequencing. RNA was prepared from areas with high tumour cell content macrodissected from cryosections. RNA was isolated using the NucleoSpin RNA kit (Machery&Nagel) according to the manufacturer's protocol. Sequencing libraries were prepared at the Berlin Institute of Health Sequencing Core using the standard TrueSeq stranded poly-A enrichment protocol. Libraries were sequenced on an Illumina HiSeq sequencer as 2 × 75nt paired-end reads. After de-multiplexing, raw read sequences in Fastq format were aligned to the human genome (assembly hg38) and *Gencode transcript annotation* (version 26)[66] using *STAR alignment* software[67]. Gene expression was determined as read counts per gene using the *featureCounts* tool in *subread* software. All further analyses of read-count data were performed using the *DESeq2* Bioconductor/R package[68]. The list of the 104 genes separating neuroblastoma risk groups was generated by analysing publicly available data from 498 primary neuroblastomas[44] for differential gene expression between high- and low-risk tumours using the *limma* R package. Genes with an absolute $\log_2$-fold change >1 and a gene-wise false-discovery rate ($q$) < 0.1 were considered to be significant in this analysis and retained in the final list. Expression of these genes in our patient samples was used for the second principal component analysis.

**Detection of *MYCN* copy number by SNP array and by FISH**. For validation of WES-based estimation of *MYCN* copy numbers, a SNP array (Cytoscan HD, ThermoFisher Scientific, #901835) was performed for selected patient samples as in ref. [15]. CEL files with intensity probe signals were analysed using the *Chromosome Analysis Suite (ChAS)* software version *4.1.0.90 (r29400)* and converted to CYCHP files. Copy numbers are visualised using the allele difference plot, weighted log2 ratio and the smoothed signal. The smoothed signal is used to directly estimate copy numbers.

*MYCN* copy number was detected on FFPE tissue on single-cell level using FISH analysis. The hybridisation probe XL *MYCN* amp (MetaSystems) consists of a green-labelled probe hybridising to the *MYCN* gene region at 2p24 and an orange-labelled probe hybridising to the *NMI* gene region at 2q23 as reference. Data interpretation was done according to INRG Biology guidelines[69].

**Reporting summary**. Further information on research design is available in the Nature Research Reporting Summary linked to this article.

## Data availability

Raw sequencing data used in this study are available in the European Genome phenome archive (www.ebi.ac.uk/ega/) under the accession number EGAS00001004735 under the following dataset IDs: EGAD00001008020 (WES), EGAD00001008156 (targeted re-sequencing), EGAD00001008133 (RNA-Seq) and EGAD00010002225 (SNP array). Download of all datasets are available by contacting the study affiliated Data Access Committee (DAC) via EGA. For WES analysis, NCBI dbSNP (https://www.ncbi.nlm.nih.gov/snp) and UCSC Genome Browser (https://genome.ucsc.edu/) were used to aggregate somatic mutations. For defining cancer-related genes, public data was downloaded from COSMIC database (https://cancer.sanger.ac.uk). For MES/ADRN differentiation subsets, analyses were done based on datasets downloaded from GEO (www.ncbi.nlm.nih.gov/geo/) under accession number GSE90805. For analysis of differential gene expression between high- and low-risk tumours, datasets were downloaded from GEO with accession GSE49711.

## Code availability

Source code in R for running *sciClone* and for constructing the clonality trees is included as a zipped file in the supplementary online information (Supplementary Software 1).

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

## Acknowledgements

We thank Nicole Hübener for critical discussions and Martin Meixner for expert technical advice about WES and targeted sequencing. This work was funded by the German Ministry for Education and Research (SYSMED-NB, #01ZX1307 and #01ZX1607), the Berlin Institute of Health (TERMINATE-NB CRG04 collaborative research project) and in the TransTumVar project. A.G.H. is supported by the German Research Foundation (#398299703) and *Wilhelm Sander Stiftung*. A.G.H. and A.K. are participants in the BIH-Charité Clinician Scientist Programme. P.F.A. and I.M.A. were supported by Austrian Science Fund (FWF) Grant No. I 2799-B28. We thank the patients and their parents for granting access to tumour specimens and clinical information for this study.

## Author contributions

A.E. and J.H.S. conceived and designed the study. K.S., J.T. J.P., M.L., P.F.A., I.M.A., F.H. and A.G.H. performed experiments, conducted analyses and interpreted data. J.T., M.H., M.C.C. and R.F.S performed bioinformatic analyses and interpreted data. L.K., C.Y.C., K.H. M.B., A.S. and M.F. contributed to experiments and analyses. D.G. provided tumour specimens. S.B. performed histopathological diagnostics on tumour samples. P.H., H.E.D. and A.K. provided study material and clinical data. J.H.S., A.E., F.H. and P.H. contributed to manuscript review. K.S., J.T., A.G.H., R.F.S. and K.A. wrote the manuscript. All authors read and approved the final manuscript.

## Competing interests

The authors declare no competing interests, except for D.G. who is employed by Experimental Pharmacology and Oncology Berlin-Buch GmbH.
