## [Peer Review File · Nature Communications]

REVIEWER COMMENTS

Reviewer #1 (Remarks to the Author): Expert in neuroblastoma genomics

Schmelz and colleagues from the Eggert and Schulte group here on a variety of DNA and RNA sequencing assays from 140 distinct “spatially and temporally” distinct samples gather from 10 neuroblastoma patients (7 with high-risk disease). The major goal of this effort is stated “to address open questions about neuroblastoma intratumoral heterogeneity”, but use bulk WES and cancer gene panel to assess DNA variation and standard RNA Seq on the samples. They make the conclusion that there is continuous clonal evolution, and that clinicians might be misled by relying on a single biopsy site, which is really not a novel insight based on the published literature in this and other diseases. While the data will indeed be of interest and useful to the neuroblastoma research community, this reviewer has several methodological concerns, as well as several suggestions that might improve the readers ability to interpret the data and their conclusions.

Major concerns

1. The authors should make it clear up front in the text and in Table 1 that 7 of the 10 patients have high-risk disease. As the authors know, high-risk and non-high-risk neuroblastoma have very different genetic etiologies and mechanisms of epigenomic somatic alterations, and this needs to be considered. In addition, this reviewer had a very hard time keeping track of the number of samples from each patient and what data they are representing. For example, in Figure 1c and Table 1, subject 1003 seems to have five diagnostic samples, and five from tumor resection, but in Figure 2c, 14 and 16 separate samples are shown for each. Are WES and target capture sequencing being shown in Figure 2?

2. A very large number of samples were derived from tumor resection following chemotherapy. In this reviewer’s experience, many of these samples have very low tumor amount (thankfully), and treatment effect makes it difficult to distinguish between differentiating neuroblasts and infiltrating Schwann and other stromal cells. The authors state that only samples with >60% tumor purity was included, but do not discuss any details of how this could be so in many of the patients who must have had chemotherapy sensitive disease as they have not suffered a relapse and may be cured.

3. Visualization. I find the heatmaps in Figure 2 very difficult to interpret.

4. The authors state that there is significant diversity in SNVs and copy number alterations, but that there is transcription homogeneity. This reviewer just doesn’t understand how that could be true if the genomic diversity is of consequential biological relevance, as the authors suggest. In addition, I would recommend looking at adrenergic versus mesenchymal gene signatures which could be of particular interest in the diagnostic vs. post-chemotherapy high-risk cases. The RNA data are suspect, especially since the low-risk samples are not clearly separate from the high-risk samples in

the PCA plot in Figure 5A. Indeed, the RNA Seq analyses seems quite cursory and may have been limited by issues of tumor purity and variation in types of stroma in the sample by site.

5. The three non-high-risk cases contribute very little to the manuscript, and some broad statements about what these cases mean in terms of clinical relevance should be modified.

6. Two issues are of concern for case 1003. First, renal invasion is not true hematogenous metastatic dissemination, and this should be clarified. Second, it does appear that there was true selections for clones with extra-copies of MYCN, but 12 copies is not “high-level amplification” and needs to be measured relative to chromosome 2 centromeric genomic content, as this is more likely high-hyperdiploidy. Orthogonal methods to validate the claim of “acquisition of MYCN amplification” which generally has not been observed in my research or clinical experience. Indeed, the comment: “In line with previous reports, MYCN amplification, a hallmark of high-risk neuroblastoma, showed little variation between spatially distinct samples, but pronounced differences between time points (Fig. 1c)” seems an overstatement and to this reviewer in all other cases it was likely quite stable and needs to be referenced to another region of chromosome 2 as there likely is some variation in chr 2 ploidy.

7. Figure 4 is potentially of very high interest, but the methodology is unclear and visualization difficult to interpret. Use of SciClone for the SNVs and MEDDIC for the SCNAs is reasonable, but Figure 4 shows both SNV- and CNV-based phylogenetic trees without any attempt to reconcile. For example, 1003 shows linear evolution by SNV, but branched based on analysis of CNVs, and there is no attempt to reconcile this discrepancy. Several phylogenetic trees show the same samples on different branches (e.g. sample 18 for clones 5 and 6 in 1001, 14 and 15 for clones 6 and 7 in 1002, etc.). What is the conclusion that Figure 4 is depicting?

8. The methods for Figure 4 state: “Source code in R for running sciClone and for constructing the clonality trees is available from the authors on request.” All code used should be readily available, and not through contacting the authors.

9. The vast majority of the 931 genes on the panel are likely irrelevant to neuroblastoma. It would be important to clarify which SNVs are likely pathogenic, and enriched in hot spot mutations in oncogenes, and LOF mutations in tumor suppressor genes.

10. It is well known that high-risk neuroblastoma typically presents with widespread metastases. Figure 4g is not sufficient to make the broad statement that metastases “emerge early”.

Minor Comments

11. Introduction.

- a. Ultra-high risk as defined in this groups recent Science paper is of interest, but still a concept and not clinically implemented in any way. This should be made clear.
- b. The discussion on telomere maintenance should mention SVs and promoter mutations.
- c. The discussion of precision therapies ignores the COG MATCH and Canadian PROFYLE trials, and perhaps others.
- d. Discussion on ALK mutations does not consider fact that first and second generation ALK inhibitors were not potent enough for the majority of mutations as has been proven biochemically and clinically.
- e. Reference 8 is a lung cancer paper, and thus not sure of relevance here.

12. A VAF of 10% to be considered clonal is low. Most authors use 20% or more. This should be explained.

13. Table 1: It is not clear how the authors calculated % genome from WES data. Please clarify.

Reviewer #2 (Remarks to the Author): Expert in neuroblastoma

Schmelz et al. investigate intratumor heterogeneity (ITH) in neuroblastoma (NB). Multi-regional sampling and analysis of 10 patient tumors were performed. They show that genetic heterogeneity in mutations and chromosome aberrations occur and claim that transcriptomes are spatially homogenous. They also report clonal evolution in NB. Two cases are presented in which the current risk classification of UHR is changed depending on ITH in different samples.

Overall, this is a well performed, well written and very important work. The findings are reliable and add novelty to the research field. The findings have potential clinical implications for how NBs are diagnosed, classified into different risk groups and potentially for treatment decisions.

Major comment:

Very limited transcriptomic analyses. Only the 100 most variable genes overall have been used for the conclusion of homogenous expression over space. By this approach it is not surprising that the major differences are between tumors. Further analysis is needed.

- a) The 100 genes that were found to be most variable should be listed.
- a) It would be interesting to perform gene ontology / GSEA analysis. Specific gene sets could be investigated.
- b) Are there changes in the MES / ADRN gene signatures? Although single-cell RNA seq would be more informative it could still be of interest to analyse with existing bulk RNA-seq data.
- c) Other findings that could point to biological pathways or processes?

More analyses should also be performed over the temporal differences that were detected, that should be highlighted and discussed.

Other comments:

In general: There should be consistency regarding coloring and naming between different figures, between figure and figure text, and between main text and figures. For instance, metastasis/infiltration, ubiquitous/clonal, specific/private, non-synonymous/non-silent.

Page 3: "Indeed, neuroblastoma shows extensive genetic intratumour heterogeneity and distinct evolutionary patterns^{8,9}".

Ref. 8 is not about NB.

Page 6: "Known cancer related genes" should be clearly defined in the main text (at least how many genes that were chosen) and possibly termed "selected cancer related genes".

Page 12/13: It should be clearly stated how many cases were found to have homogenous classification and how many where the classification changed (2/10 ?).

Page 18: "...biopsies were taken from geographically separate areas..."

Please clarify minimal distance of separated tumor areas, e.g., "with minimal distance of X mm".

Figure 1:

- a) Patient ID should be better shown.
- b) Not all abbreviations are described, eg., SCNA.
- c) In the legend it says: "(b) study design and sample workflow for WES". But it seems that fig. 1b shows study design for the entire study including RNA-seq.

d) Regarding site: “metastasis” is used in the main text and “infiltration” in the figure. Please clarify if it is distant metastasis or local infiltration and use consistent terminology.

Figure 2:

a) BRCA2 mutations are not mentioned in text. Why?

b) Why are not all diagnostic samples (shown in b and c) included in a)

c) b) and c) do not add much information. It is difficult to see the red borders. This should be displayed in a better way if included.

d) PDXs samples are suddenly included. PDXs are however not mentioned in the main text or in Materials&Methods. Why are they included? Do they contribute in anyway? Consider removing from the manuscript if they do not contribute.

e) Again, please clarify if it is distant metastasis or local infiltration and use consistent terminology.

Figure 3:

a) This it is a bit confusing. The y-axis denotes the number of patients with each aberration? If so, it should not be -5 on the y-scale.

b) MSAI should be explained in the main text and not only referred to.

Figure 4:

a) Increase the visibility of the legend with the colors

b) The legend does not include all colors, please include and explain all distinct colors

c) Avoid redundancy: The SNV tree for patient 1001 is shown in both a) and b)

d) It might be difficult to follow fig. 4 c-g. Please clarify.

Daniel Bexell

Reviewer #3 (Remarks to the Author): Expert in neuroblastoma computational genomics and evolution

Schmelz and colleagues present a spatiotemporal genomic analysis of 10 neuroblastoma patients, using multi-site exome and transcriptome sequencing. The samples from diagnosis, metastasis and resection from the same patient may be profiled, allowing construction of evolution tree using somatic point mutations and copy number alterations (SCNA). Interesting observations include 1) heterogeneous distribution of targetable variations at diagnosis and relapse, raising concerns on whether single biopsies is sufficiently reliable for therapy decisions; 2) early emergence of metastatic clones and ongoing chromosomal instability during disease evolution; and 3) changes of tumor risk stratification based on transcription analysis. These observations are important as the data can support changes in neuroblastoma clinical practice such as performing multiple biopsy which may improve therapy options for patients.

A few major and minor comments are listed below:

1. Most of the genes reported as potential drivers in Fig. 2a, based on COSMIC, are not credentialed neuroblastoma driver genes and are most likely passengers. Based on the description in Methods, non-synonymous variants in COSMIC Cancer Gene Census are broadly referred as “driver” or “pathogenic variants”. Such a loose definition may lead to non-synonymous passenger variants being misclassified as driver. For example, COL1A1 is listed in the COSMIC Cancer Gene Census because it is fused to PDGFB in some cancer types. Missense mutations, like those reported in the study by Schmelz and colleagues, are not reported as COSMIC driver variants. NOTCH mutations are functional primarily in hematopoietic lineages, and aren’t known drivers in neuroblastoma. The KIF1B and DPYSL5 variants that are highlighted are not known neuroblastoma drivers and this should be noted in the text unless the authors know of a reference indicating its driver status in neuroblastoma. Knowing the protein site localization, such as whether the FGFR1 variant is a hotspot, and whether the mutations are expressed or not would also give hints as to their driver status. The known neuroblastoma driver genes should be noted in some way, such as in a separate color, to make it clear which are likely drivers.

2. The heatmaps in Fig. 2 are difficult to read and interpret due to the black background and the counter-intuitive use of lighter colors to show higher VAF. Inverting the color scheme would likely make these easier to interpret. For selected variants of special interest, such as ALK, FGFR1, etc., it would be helpful to show a bar or line plot with VAF on one axis and sample on the other, so that the VAFs can be assessed numerically rather than by color.

3. Fig. 3 shows a valuable “average” summary across the cohort. It would also be helpful to show a few heatmaps of copy profiles, where each heatmap shows a specific patient having copy number heterogeneity. This would make it easier to evaluate and interpret the reported copy number heterogeneity within a specific patient. Panel B can be moved to Supplementary as it is depicting a well-known knowledge that whole-genome duplication is more common in low-risk patients.

4. Distinct evolution trajectory of metastatic clone presented in Figure 4 is very interesting; however, the authors reached conclusion that metastatic clones were originated from early evolution

branches (stated in abstract and results) based on the following data: “Both SCNA and SNV data indicated early branching of the metastatic clone, with clear evidence of clonal aberrations within the metastasis that were not present in the primary tumour (Fig. 4b, Supplementary Fig. S2a).” It should be noted that the argument for early evolution can only be inferred when the clonal variants in primary tumors are absent in metastasis but not the other way around. It will make more sense to place metastatic sample in parallel with the resected samples or as “descendants” of the diagnostic samples in Figure 4b, e. This also raised the question on the criteria used for constructing evolutionary tree using SCNA data.

5. Does the UHR classification heterogeneity section (Fig. 5b) rely entirely on the fact that patients 1003 and 1008 had heterogeneity in MYCN and ALK alterations? The authors need to comment on that.

6. Figure 4b and e shown clonal structure derived from SCNV analysis. It might be good to combine this together with the relevant graph in 4a so that it is clear the SCNV analysis is consistent with those from SNV/indel. Supplementary Fig. S3 should also be modified to mark the CNV status integrated with SNV/indel. Figure 4f: sample 21 (metastatic sample) has the same logR profile (for CNV amplitude) as sample 23 (resected) but lacks the allelic imbalance shown in sample 23. It should be noted that according to text on page 11, sample 23 should have 1-copy gain (“all samples obtained from the resected tumour showed clear allelic imbalance and a single-copy gain of chromosome arm 9q, which was absent in both the diagnostic biopsies and the metastasis resected synchronously with the primary tumour (Fig. 4g)”). Figure 4b shown that sample 21 (metastatic sample) has the same amplitude of CNV (logR graph, 1-copy change) as sample 23 (resected sample), indicating a potential error in Figure 4d regarding the logR ratio of sample 21. The authors need to verify their data and confirm the copy number variation status of sample 21. If sample 21 is indeed having a logR comparable to sample 23 and lacks allelic imbalance, this can represent a subclonal two-copy gain (duplication of both haplotypes) and should be noted in the text or figure legend.

7. In the paper, two different evolution schemes are used: (a) the “ubiquitous/shared/private” scheme and (b) the “clonal/subclonal” scheme. Since these two schemes convey closely related information, it would be helpful to clarify to readers why both schemes were discussed, or alternatively, to use only one or the other. The criteria for defining clonal appears to be better suited for the definition of ubiquitous as it only requires a VAF of 0.1 (i.e. “SNVs were considered to be clonal if present with a variant allele frequency (VAF) greater than 10% in WES or targeted sequencing data from all samples. All other detected SNVs were considered subclonal”).

8. In patient CB1008, was the loss of the ALK variant after treatment due to a different anatomical site being sampled? Or was the same anatomical location sequenced in both the pre-treatment and post-treatment samples? This would indicate whether the change was related to temporal vs. spatial evolution. In general, indicating the specific site of each sample (e.g. adrenal, paraspinal, distant met location, etc.) in, for example, Fig. 1, would be useful to evaluate spatial vs. temporal differences.

9. Page 7 “Spatial and temporal heterogeneity of single-nucleotide variants affects therapeutically actionable genes in neuroblastoma”. Whereas between two and six non-synonymous SNVs were detected in known cancer-related genes (defined by the COSMIC database²⁰ and other studies^{6,10,21}) in high-risk tumours, very few or no non-synonymous SNVs were detected in neuroblastomas from other risk groups (Fig. 2a).

Is this related to the overall higher mutational burden in high risk samples compared with the low risk samples.

Minor concerns

1. The paper is well-written overall, but a few statements were unclear. For example, the statement “Focal amplifications of the MYCN oncogene, re-arrangements in the nearby TERT locus on chromosome 5p...” was confusing since MYCN and TERT are not nearby, but on different chromosomes. Additionally, the statement “On average, between 13-92% (median 34%, mean 42%) of each genome was affected by at least one SCNA event in at least one sample (Fig. 3a)” was a bit hard for me to follow. Does “each genome” refer to “each sample”? The terms “diagnosis of relapse” in the text and “relapse diagnosis” in Table 1 are also a bit ambiguous.

2. Figure 2C/B: status of WES, FFPE or PDX can be combined into one bar to save space. Yellow grid plus the red lines makes it hard to discern the blue shade used for depicting the VAF. BRCA2 appeared twice in the same sample. Is this an error in data presentation?

3. Gains of 2p and 7q were observed in all or almost all of the 10 patients reported. This is higher than expected, based on the authors’ references 6 and 16, for example. Is there some selection criteria for this study that might explain this?

4. The trees in Fig. 4a and 4b were difficult for me to interpret. In particular, two numbering schemes are used, with (1) numbers inside the circles at the end of tree branches, and (2) a separate numbering system with (samples?) numbered in colors based on diagnosis/resection/relapse/metastasis status. Do these different numbering systems refer to mutation clusters vs. samples, respectively? Are these clone trees or sample trees? It’s difficult to tell from the legend and clarification would be helpful.

5. From Supplementary Fig. 1a, it appears that more samples were subjected to targeted sequencing than WES. Were both WES and targeted sequencing performed on some samples, while other samples received targeted sequencing only? If yes, wouldn’t the samples with targeted sequencing

only lack any private variants since de novo detection of variants wasn't performed in these samples? Would this affect or confound any of the tree structures shown in Figs. 4a, b (i.e. were the targeted sequencing-only samples without WES included in these figures)?

6. Does the transcriptional change reported in CB1003 correspond to a shift between the two neuroblastoma transcriptional differentiation states reported in van Groningen et al., Nature Genetics 2017? If not, what specific transcriptional differences are seen before vs. after treatment? The transcriptional evolution of this patient, and the transcriptional differences between high- and low-risk patients, are of interest and it would be helpful to clarify what specific genes or gene signatures are variable in these settings.

7. The Supplementary Figure 2 text to indicate panels "f" and "h" appear to be missing.

8. Figure 5a needs to mark the low & high risk tumor groups

Point-by-point response to reviewer comments for manuscript NCOMMS-20-46778

Reviewer #1 (Remarks to the Author): Expert in neuroblastoma genomics

Schmelz and colleagues from the Eggert and Schulte group here on a variety of DNA and RNA sequencing assays from 140 distinct “spatially and temporally” distinct samples gather from 10 neuroblastoma patients (7 with high-risk disease). The major goal of this effort is stated “to address open questions about neuroblastoma intratumoral heterogeneity”, but use bulk WES and cancer gene panel to assess DNA variation and standard RNA Seq on the samples. They make the conclusion that there is continuous clonal evolution, and that clinicians might be misled by relying on a single biopsy site, which is really not a novel insight based on the published literature in this and other diseases. While the data will indeed be of interest and useful to the neuroblastoma research community, this reviewer has several methodological concerns, as well as several suggestions that might improve the readers ability to interpret the data and their conclusions.

Major concerns

1. The authors should make it clear up front in the text and in Table 1 that 7 of the 10 patients have high-risk disease. As the authors know, high-risk and non-high-risk neuroblastoma have very different genetic etiologies and mechanisms of epigenomic somatic alterations, and this needs to be considered. In addition, this reviewer had a very hard time keeping track of the number of samples from each patient and what data they are representing. For example, in Figure 1c and Table 1, subject 1003 seems to have five diagnostic samples, and five from tumor resection, but in Figure 2c, 14 and 16 separate samples are shown for each. Are WES and target capture sequencing being shown in Figure 2?

Response: We thank the reviewer for these very helpful suggestions and regret that the manuscript was not clear enough. We included the information of the number of high-risk patients in lines 97 to 99 and in Table 1. We made several changes in the revised manuscript to better track patient samples and subsequent analysis. We clarified in lines 108 to 110 that we performed targeted sequencing using a panel design on the basis of 1476 SNVs detected by WES in 51 snap-frozen samples. Following WES analysis of 51 samples, we included a further 89 samples to total 140 samples analyzed by ultra-deep targeted sequencing in the revised manuscript. We also changed Fig. 2 accordingly.

2. A very large number of samples were derived from tumor resection following chemotherapy. In this reviewer’s experience, many of these samples have very low tumor amount (thankfully), and treatment effect makes it difficult to distinguish between differentiating neuroblasts and infiltrating Schwann and other stromal cells. The authors state that only samples with >60% tumor purity were included, but do not discuss any details of how this could be so in many of the patients who must have had chemotherapy sensitive disease as they have not suffered a relapse and may be cured.

Response: We appreciate the questions raised by the reviewer and are thankful for pointing this out. Each sample underwent stringent quality control performed on H&E-stained cryosections or FFPE sections of tumor regions. Only tumor regions with high tumor cell content (>60% for WES and RNA Seq, >10% for targeted seq) were included in further analysis. A pathologist (S.B.) confirmed the content of vital tumor cells, evaluated the rate of differentiation and the amount of stroma from H&E stained sections and marked appropriate areas which were then macrodissected on sequential cryosections or FFPE sections. This is now described in more detail in the methods section (lines 477 to 479) and is included in the results (lines 103 to 105). In addition, the expression level of neuroblastoma markers such as PHOX2B and GATA2 was used as supporting information for sufficient tumor cell content, and 2 samples were excluded from the analysis due to very low marker expression.

3. *Visualization. I find the heatmaps in Figure 2 very difficult to interpret.*

Response: We very much appreciate this comment from the reviewer and have thoroughly changed the coloring in heatmaps in Fig. 2 and the Supplementary Fig. 2a-h. We have switched the colors representing VAF (now: high VAF as dark blue, low VAF as light blue on a white background) to make the heatmaps more intuitive, clearer and easier to interpret. We have also added a bar plot displaying the VAF of selected cancer-related genes for the 2 patients in Fig. 2b and c (Supplementary Fig. 2i+j).

4. *The authors state that there is significant diversity in SNVs and copy number alterations, but that there is transcription homogeneity. This reviewer just doesn't understand how that could be true if the genomic diversity is of consequential biological relevance, as the authors suggest. In addition, I would recommend looking at adrenergic versus mesenchymal gene signatures which could be of particular interest in the diagnostic vs. post-chemotherapy high-risk cases. The RNA data are suspect, especially since the low-risk samples are not clearly separate from the high-risk samples in the PCA plot in Fig. 5A. Indeed, the RNA Seq analysis seems quite cursory and may have been limited by issues of tumor purity and variation in types of stroma in the sample by site.*

Response: We appreciate the reviewer's suggestions and agree that it is worthwhile and necessary to analyze the RNA expression data in more detail. Please also see our responses to reviewer #2 major comments a) to d) and reviewer #3 point 6. To address this point, we assessed the impact of the most prominent copy number changes in the cohort, which was the *MYCN* amplification. *MYCN* copies clearly correlated with gene expression of *MYCN* and of its target gene, *TERT*. In contrast, samples from CB1002, CB1004 and CB1010 displayed high *TERT* expression, but lacked *MYCN* amplifications. These samples were also negatively tested for *TERT* rearrangement by break-apart FISH (data not shown). These findings are in line with Ackermann et al, who reported high *TERT* expression of unknown origin in a subset of neuroblastomas¹ (Supplementary Fig. 6a). Other SCNAs clearly correlated with changes in gene expression, such as the gain in chr 7, which was detected in samples from 8 of 9 patients, and was correlated with heightened transcription from *IL6*, located on chr. 7 *p15.3*. This is now outlined in the revised manuscript on lines 300 to 315 and shown in Supplementary Fig. 6a.

We thank the reviewer for the recommendation to look at adrenergic versus mesenchymal gene signatures in diagnostic versus post-chemotherapy high-risk cases. We performed an analysis using gene signature scores from Groningen et al.² for patient samples from CB1003 and CB1008, but found no significant change in differentiation state signatures between the two time points (Supplementary Fig. 6c+d). This analysis might be limited based on bulk RNA Seq data and would be more informative if applied to single-cell data, which could discriminate between tumor and stromal cells.

We also re-performed the PCA based on risk-associated gene expression from Fig. 5 after thoroughly checking for neuroblastoma markers to verify comparable tumor purity among samples from one patient (Fig. 5). For more clarity, we also reorganized the figure and marked tumor groups associated with high and low risk. The revised manuscript contains an updated Fig. 5a, where the first and second principal components clearly separate the high-risk samples from the low- and intermediate-risk samples at the zero points, which is now visually supported by arrows beneath the axis. The shift of CB1003 and CB1008 to different risk-associated tumor groups is highlighted by arrows within the plot.

5. The three non-high-risk cases contribute very little to the manuscript, and some broad statements about what these cases mean in terms of clinical relevance should be modified.

Response: We thank the reviewer for this comment. With due respect, we are not aware of any broad statement about the low- and intermediate-risk neuroblastomas in our cohort with respect to overall survival. We reported the finding that samples from these cases harbored a significantly lower number of non-silent mutations compared to the high-risk neuroblastomas, which is also in line with previous reports. These references are now included in the revised manuscript (line 118)^{1,3,4}.

6. Two issues are of concern for case 1003. First, renal invasion is not true hematogenous metastatic dissemination, and this should be clarified. Second, it does appear that there was true selections for clones with extra-copies of MYCN, but 12 copies is not “high-level amplification” and needs to be measured relative to chromosome 2 centromeric genomic content, as this is more likely high-hyperdiploidy. Orthogonal methods to validate the claim of “acquisition of MYCN amplification” which generally has not been observed in my research or clinical experience. Indeed, the comment: “In line with previous reports, MYCN amplification, a hallmark of high-risk neuroblastoma, showed little variation between spatially distinct samples, but pronounced differences between time points (Fig. 1c)” seems an overstatement and to this reviewer in all other cases it was likely quite stable and needs to be referenced to another region of chromosome 2 as there likely is some variation in chr 2 ploidy.

Response: We thank the reviewer for this comment and have clarified this misunderstanding in the revised manuscript. The metastasis from patient CB1003 is not a renal infiltration, but a lymph node metastasis. Renal invasion occurred in one sample from patient CB1002, and the probable way of metastasizing has been included in the

text (lines 255 to 256). We had actually labeled the *MYCN* amplification as “12+” in Fig. 4. The actual estimates from WES are 16 in the 4 regions from the second look, and we have corrected the figure accordingly. The defining threshold for amplification used here is a copy-number greater than twice the average tumor ploidy, a common definition of amplification (see e.g. Watkins et al. 2020, Nature)⁵. We agree that the term “high-level” might not be warranted, and removed it from the section.

We thank the reviewer for the very valuable suggestion regarding the validation of the *MYCN* amplification. We have now performed a quantification of the FISH analysis of diagnostic and resection samples from CB1003, comparing signals for *MYCN* and the long arm of chromosome 2. We have also added data from a SNP array as an orthogonal method to validate acquisition of the *MYCN* amplification in the tumor resection samples from CB1003. This is now included in a revised Supplementary Fig. 5. We have specified the statement about the temporal *MYCN* heterogeneity in CB1003 and changed the text accordingly (line 129). In addition, we leveraged RNA-seq data to validate the claim that *MYCN* undergoes amplification at the second time point (compared to diagnosis). Supplementary Fig. 6 shows a clear correlation between *MYCN* expression and CN as estimated from the WES data and a clear separation between time points for CB1003, in line with the late amplification of *MYCN*.

7. Figure 4 is potentially of very high interest, but the methodology is unclear and visualization difficult to interpret. Use of SciClone for the SNVs and MEDDIC for the SCNAs is reasonable, but Figure 4 shows both SNV- and CNV-based phylogenetic trees without any attempt to reconcile. For example, 1003 shows linear evolution by SNV, but branched based on analysis of CNVs, and there is no attempt to reconcile this discrepancy. Several phylogenetic trees show the same samples on different branches (e.g. sample 18 for clones 5 and 6 in 1001, 14 and 15 for clones 6 and 7 in 1002, etc.). What is the conclusion that Figure 4 is depicting?

Response: We are glad to hear that the reviewer appreciates the clonality analysis in Fig. 4 and thank the reviewer for the comments. We have added a comparison of the SNV- and SCNA-based trees for CB1003. We thoroughly revised the manuscript and added more detailed information to the results in Fig. 4 to better enable interpretation of the data and a conclusion to be drawn from the phylogeny, which is added in more detail to the text (lines 239 to 244 and 255 to 269). We were also intrigued by the discrepancy between the SNV and CNV trees for CB1003. We re-analyzed all copy number-based phylogenies using MEDICC2⁶, an improved version of MEDICC that also allows for assessing robustness of the trees. While all other topologies remained as before, MEDICC2 inferred a new topology for CB1003 with branch lengths more similar to what is evident from the SNV analysis. The diagnostic samples now have a substantially shorter branch at their ancestral clade compared to the tumor resection samples, indicating a more “linear” evolutionary trajectory in line with the SNV phylogenies. To improve overall comparison between SNV and SCNA trees, we have extended Supplementary Fig. 4 with the new data that includes copy number tracks for each sample and reconstructed ancestral genomes. Despite these similarities, there is a fundamental difference in the way SNV- and SCNA-based trees are constructed, in that

SNV inference algorithms attempt clonal deconvolution for each sample and try to build clone trees. Since multiple clones can co-exist in a single sample, samples can occur at multiple positions in the SNV tree. The SCNA algorithms, in contrast, extract the major or dominant clone per sample and use this for phylogenetic inference, so that samples occur only once. Despite this technical difference, we have repeatedly done comparative analyses between SCNA- and SNV-based trees in the context of other research projects, and found that they are typically in good agreement across cancer types (see, for example ⁶) since almost always one clone clearly dominates a given sample.

8. *The methods for Figure 4 state: “Source code in R for running sciClone and for constructing the clonality trees is available from the authors on request.” All code used should be readily available, and not through contacting the authors.*

Response: We thank the reviewer for this comment. The scripts used to perform sciClone analysis have been included in the online supplementary information as a zip.file.

9. *The vast majority of the 931 genes on the panel are likely irrelevant to neuroblastoma. It would be important to clarify which SNVs are likely pathogenic, and enriched in hot spot mutations in oncogenes, and LOF mutations in tumor suppressor genes.*

Response: We very much appreciate this suggestion from the reviewer. We have revised the panel of 931 cancer-related genes according to a more stringent definition. We have excluded genes with only a putative relevance for cancer (tier 2 in the COSMIC database) and restricted the table to genes with proven relevance for cancer (tier 1 of COSMIC) in addition to aberrations relevant for pediatric cancers (Gröbner et al., Worst et al.) ^{7,8} or neuroblastoma-relevant aberrations (Pugh et al., Ackermann et al., Brady et al.) ^{1,3,4}. This compilation of cancer-related genes now comprises 801 genes, and is added as Supplementary File 1. All affected figures have been changed accordingly. We agree that the relevance for neuroblastoma has only been shown for a limited number of gene mutations. We have changed Fig. 2a accordingly and marked neuroblastoma-related mutations in red based on the aforementioned publications on neuroblastoma genetics. We have also included a table (Supplementary Table 2) providing information of the type and putative biological consequences of all the cancer-related SNVs in Fig. 2a.

10. *It is well known that high-risk neuroblastoma typically presents with widespread metastases. Figure 4g is not sufficient to make the broad statement that metastases “emerge early”.*

Response: We agree that “early” is not a precise statement, and since this was pointed out by other reviewers as well, we have adjusted the phrasing in the corresponding sections. In particular, the section on the SCNA phylogenies is now called “Inference of evolutionary trajectories allows relative timing of the emergence of metastases in neuroblastoma evolution” and we discuss both cases, CB1001 and CB1003, with respect to metastatic dissemination in more detail (see also comment #7 above).

Minor Comments

11. *Introduction.*

a. *Ultra-high risk as defined in this groups recent Science paper is of interest, but still a concept and not clinically implemented in any way. This should be made clear.*

Response: We changed the text accordingly in line 50.

b. *The discussion on telomere maintenance should mention SVs and promoter mutations.*

Response: We thank the reviewer for this comment, and added these causes for high TERT expression, and accordingly, telomere maintenance in line 51.

c. *The discussion of precision therapies ignores the COG MATCH and Canadian PROFYLE trials, and perhaps others.*

Response: We thank the reviewer for this remark. We included the MATCH and PROFYLE study. We also included a paper reviewing recent efforts on precision therapies for pediatric malignancies (lines 54, 55).

d. *Discussion on ALK mutations does not consider fact that first and second generation ALK inhibitors were not potent enough for the majority of mutations as has been proven biochemically and clinically.*

Response: We thank the reviewer for this comment. We included two references in the revised manuscript on ALK inhibitor resistance due to ineffectiveness of the 1st and 2nd generation compounds against activating mutations^{9,10}.

e. *Reference 8 is a lung cancer paper, and thus not sure of relevance here.*

Response: We have exchanged ref 8. We replaced the paper focusing on lung cancer (Jamal-Hanjani et al. 2017) with one focusing on neuroblastoma¹¹ (line 61).

12. *A VAF of 10% to be considered clonal is low. Most authors use 20% or more. This should be explained.*

Response: We thank the reviewer for this suggestion. We explained our decision to avoid overestimation of heterogeneity in the revised manuscript (line 121).

13. *Table 1: It is not clear how the authors calculated % genome from WES data. Please clarify.*

Response: Allele-specific SCNA calling is based on SNP positions (see e.g. Methods). While the reviewer is, of course, right that the SNP density is much lower in WES data compared to WGS data, the number of SNPs available is a sufficiently large to create chromosome-spanning copy-number segmentations. While it is possible that very small copy-number events might escape us using WES data, such as an event only affecting intergenic regions, these segments will still be small compared to the overall segment size and will not drastically affect the estimated proportions. In other words, while the variance of the estimator is certainly higher in WES data compared to WGS, there should not be a systematic bias. For details on the genome-wide segmentation of copy-number states, please also see the revised Supplementary Figure 4, which contains all segmentation data for the cohort.

Reviewer #2 (Remarks to the Author): Expert in neuroblastoma

Schmelz et al. investigate intratumor heterogeneity (ITH) in neuroblastoma (NB). Multi-regional sampling and analysis of 10 patient tumors were performed. They show that genetic heterogeneity in mutations and chromosome aberrations occur and claim that transcriptomes are spatially homogeneous. They also report clonal evolution in NB. Two cases are presented in which the current risk classification of UHR is changed depending on ITH in different samples.

Overall, this is a well performed, well written and very important work. The findings are reliable and add novelty to the research field. The findings have potential clinical implications for how NBs are diagnosed, classified into different risk groups and potentially for treatment decisions.

Major comment:

Very limited transcriptomic analyses. Only the 100 most variable genes overall have been used for the conclusion of homogenous expression over space. By this approach it is not surprising that the major differences are between tumors. Further analysis is needed.

Response: We very much appreciate the suggestions of the reviewer and agree that a more detailed RNA-Seq analysis is warranted. Following these suggestions, we performed various analyses as listed below that confirmed the temporal intratumor heterogeneity of gene expression in accordance with the temporal appearance of SCNA. Please also see our responses to reviewer #1, point 4.

a) The 100 genes that were found to be most variable should be listed.

Response: We included the list of the 100 most variable genes as Supplementary file 5.

b) It would be interesting to perform gene ontology / GSEA analysis. Specific gene sets could be investigated.

Response: We thank the reviewer for these suggestions and performed a GSEA of *MYCN*-regulated genes based on a gene set list included in the revised submission as Supplementary file 3. GSEA focused on these patients CB1003 and CB1008, since temporal SCNA intratumor heterogeneity was most prominent for the *MYCN* locus in these patient samples. *MYCN*-regulated genes were enriched towards the CB1003 tumor resection, which is in line with acquisition of the *MYCN* amplification at that time point. In CB1008, the gene set of *MYCN*-regulated genes was significantly enriched towards diagnosis, which agrees with the decrease of *MYCN* copy number during tumor

progression. The revised version has this data included in Supplementary Fig. 6 and outlined in the revised manuscript in lines 300 to 315.

c) Are there changes in the MES / ADRN gene signatures? Although single-cell RNA seq would be more informative it could still be of interest to analyze with existing bulk RNA-seq data.

Response: We thank the reviewer for the recommendation to look at adrenergic versus mesenchymal gene signatures. Please also see our responses to reviewer #1 point 4 and reviewer #3 point 6. We performed an analysis using gene signature scores as in ² for CB1003 and CB1008 patient samples, but found no significant change in differentiation state signatures between the two time points (Supplementary Fig. 6). We fully agree with the reviewer that this analysis might be limited based on bulk RNA-Seq data, and would be more informative using single-cell data, which could discriminate between tumor and stromal cells. In conclusion, we did not find significant changes in differentiation states at diagnosis versus tumor resection, but could not fully exclude its presence due to limited resolution of our bulk RNA expression data analysis.

d) Other findings that could point to biological pathways or processes?

More analyses should also be performed over the temporal differences that were detected, that should be highlighted and discussed.

Response: We thank the reviewer for pointing this out. We performed a gene ontology analysis for RNA sequencing data for longitudinal samples available from the patients CB1003, CB1008 and CB1009. Among the top 20 pathways, this detected “cytokine pathogenesis” and “cytokine therapy”. Marker genes for activity in these pathways were further analyzed for changes over time. As an example, significantly altered gene expression between two time points were shown for CB1003 samples (Supplementary Fig. 6f). In tumor resection samples from CB1003, but not from CB1008 or CB1009, we detected a significant increase in *CCL2*, *HIF1A* and *IFNG* expression, suggesting that the therapy had a different immunomodulatory effect among patients.

Other comments:

In general: There should be consistency regarding coloring and naming between different figures, between figure and figure text, and between main text and figures. For instance, metastasis/infiltration, ubiquitous/clonal, specific/private, non-synonymous/non-silent.

Response: We thank the reviewer for these very helpful suggestions, and regret that the manuscript lacked consistency in coloring and naming. We now thoroughly adjusted definitions and naming in the revised manuscript. We differentiated more exactly between metastasis and infiltration and indicated this in the revised manuscript text and figures. We also harmonized “ubiquitous/clonal” to clonal according to our definition of “present in all samples from a patient with a VAF < 10%”. We decided for “specific” and for “non-silent” and changed the manuscript accordingly.

Page 3: "Indeed, neuroblastoma shows extensive genetic intratumour heterogeneity and distinct evolutionary patterns 8,9".

Ref. 8 is not about NB.

Response: We thank the reviewer for this comment, and have exchanged the previously referenced paper focusing on lung cancer (Jamal-Hanjani et al. 2017) as ref 8 for a neuroblastoma-related paper ¹¹ (line 63).

Page 6: “Known cancer related genes” should be clearly defined in the main text (at least how many genes that were chosen) and possibly termed “selected cancer related genes”.

Response: We very much appreciate the suggestion of the reviewer. (See also our responses to reviewer #1) We have revised the panel of 931 cancer-related genes according to a more stringent definition. We have excluded genes with only a putative relevance for cancer (tier 2 in the COSMIC database) and restricted the table to genes with proven relevance for cancer (tier 1 of COSMIC) in addition to aberrations that are relevant for pediatric cancers (Gröbner et al., Worst et al.) ^{7,8} or neuroblastoma (Pugh et al., Ackermann et al., Brady et. al) ^{1,3,4}. This compilation of cancer-related genes now consists of 801 genes (line 137) and is added to the revised supplement as Table 2, to which the revised text refers to in line 139. All affected figures have been changed accordingly. We also included Supplementary table 2, which provides information about the type and putative biological consequences of all the cancer-related SNVs from Fig. 2a.

Page 12/13: It should be clearly stated how many cases were found to have homogenous classification and how many where the classification changed (2/10?).

Response: Thank you for pointing this out. We included this information in the revised text in lines 347 to 348.

Page 18: “...biopsies were taken from geographically separate areas...”

Please clarify minimal distance of separated tumor areas, e.g., “with minimal distance of X mm”.

Response: Thank you for your suggestion. We changed the text accordingly (line 475).

Figure 1:

a) Patient ID should be better shown.

Response: Thank you for your suggestion. We changed Fig. 1 accordingly.

b) Not all abbreviations are described, eg., SCNA.

Response: Thank you for your comment. We changed the legend to Fig. 1 accordingly.

c) In the legend it says: “(b) study design and sample workflow for WES”. But it seems that fig. 1b shows study design for the entire study including RNA-seq.

Response: Thank you for your suggestion. We changed the legend to Fig. 1 accordingly.

d) Regarding site: “metastasis” is used in the main text and “infiltration” in the figure. Please clarify if it is distant metastasis or local infiltration and use consistent terminology.

Response: We thank the reviewer for this comment. In the revised manuscript (including figures), we now precisely differentiate between distant metastasis and metastatic infiltration.

Figure 2:

a) BRCA2 mutations are not mentioned in text. Why?

Response: We thank the reviewer for this question. We captured two different SNVs in the regulatory upstream region of *BRCA2* (flanking intronic regions) by WES. The biological relevance of these SNVs is unclear, therefore, we did not mention them in the text but listed all their details in the cancer-related genes and their putative consequences in Supplementary Table 2.

b) Why are not all diagnostic samples (shown in b and c) included in a)

Response: Thank you very much for this comment. We regret that the workflow was not clearly presented. Fig. 2a shows data from 51 samples (from the 10 patients) which were exclusively analyzed by WES. Based on this, 1479 SNVs were selected for the panel design of a targeted re-sequencing in 89 further samples including FFPE material and material with lower tumor purity. Fig. 2a and b show targeted re-sequencing results for all available samples from patient CB1002 (b) and CB1003 (c), including samples that had already been analyzed by WES and additional samples analyzed only by targeted sequencing. To avoid further misunderstanding, this is now indicated in Fig. 2 and outlined in the manuscript (lines 108 to 110). See also our response to reviewer #1.

c) b) and c) do not add much information. It is difficult to see the red borders. This should be displayed in a better way if included.

Response: We much appreciate your suggestions, and revised the graphical design of Fig. 2. See also our response to reviewer #1

d) PDXs samples are suddenly included. PDXs are however not mentioned in the main text or in Materials&Methods. Why are they included? Do they contribute in anyway? Consider removing from the manuscript if they do not contribute.

Response: Thank you for your suggestions. We agree that they do not substantially contribute and excluded the PDX data from Fig. 2c.

e) Again, please clarify if it is distant metastasis or local infiltration and use consistent terminology.

Response: Thank you for this comment. We specified metastasis and local infiltration in Fig. 2 and in the revised manuscript.

Figure 3:

a) This it is a bit confusing. The y-axis denotes the number of patients with each aberration? If so, it should not be -5 on the y-scale.

Response: Thank you for this comment. The y-axis gives the number of the patients with gains (in red) as upward bars and the number of patients with losses (blue) as downward bars. We agree that the annotation is misleading and changed it into positive scales on the y-axis.

b) MSAI should be explained in the main text and not only referred to.

Response: We thank the reviewer for this suggestion. MSAI is now explained in lines 187 to 188 in the revised manuscript.

Figure 4:

a) Increase the visibility of the legend with the colors

Response: We thank the reviewer for this suggestion. We changed the size of the legend for better visibility.

b) The legend does not include all colors, please include and explain all distinct colors

Response: Thank you for this comment. We explained the remaining colors (of the different tree branches) in the revised figure legend.

c) Avoid redundancy: The SNV tree for patient 1001 is shown in both a) and b)

Response: We thank the reviewer for this suggestion. We excluded the panel for patient CB1001 in Fig. 4a.

c) It might be difficult to follow fig. 4 c-g. Please clarify.

Response: We agree that the figure was difficult to follow. We completely re-worked Fig. 4, providing a side-by-side comparison of SNV, SCNA trees and the corresponding SCNA changes for patients CB1001 and CB1003. We also updated Supplementary Fig 4 which compares SNV and SCNA trees for all patients. We hope that Fig. 4 is clearer now (including the legend etc.).

Reviewer #3 (Remarks to the Author): Expert in neuroblastoma computational genomics and evolution

Schmelz and colleagues present a spatiotemporal genomic analysis of 10 neuroblastoma patients, using multi-site exome and transcriptome sequencing. The samples from diagnosis, metastasis and resection from the same patient may be profiled, allowing construction of evolution tree using somatic point mutations and copy number alterations (SCNA). Interesting observations include 1) heterogeneous distribution of targetable variations at diagnosis and relapse, raising concerns on whether single biopsies is sufficiently reliable for therapy decisions; 2) early emergence of metastatic clones and ongoing chromosomal instability during disease evolution; and 3) changes of tumor risk stratification based on transcription analysis. These observations are important as the data can support changes in neuroblastoma clinical practice such as performing multiple biopsy which may improve therapy options for patients.

A few major and minor comments are listed below:

1. Most of the genes reported as potential drivers in Fig. 2a, based on COSMIC, are not credentialed neuroblastoma driver genes and are most likely passengers. Based on the description in Methods, non-synonymous variants in COSMIC Cancer Gene Census are broadly referred as “driver” or “pathogenic variants”. Such a loose definition may lead to non-synonymous passenger variants being misclassified as driver. For example, COL1A1 is

listed in the COSMIC Cancer Gene Census because it is fused to PDGFB in some cancer types. Missense mutations, like those reported in the study by Schmelz and colleagues, are not reported as COSMIC driver variants. NOTCH mutations are functional primarily in hematopoietic lineages and aren't known drivers in neuroblastoma. The KIF1B and DPYSL5 variants that are highlighted are not known neuroblastoma drivers and this should be noted in the text unless the authors know of a reference indicating its driver status in neuroblastoma. Knowing the protein site localization, such as whether the FGFR1 variant is a hotspot, and whether the mutations are expressed or not would also give hints as to their driver status. The known neuroblastoma driver genes should be noted in some way, such as in a separate color, to make it clear which are likely drivers.

Response: We very much appreciate these suggestions from the reviewer and would like to refer to our responses to reviewer #1, point 9 and reviewer #2, major comment point c).

We have revised the panel of 931 cancer-related genes according to a more stringent definition. We have excluded genes with only a putative relevance for cancer (tier 2 in the COSMIC database) and restricted the table to genes with proven relevance for cancer (tier 1 of COSMIC) and relevant aberrations for pediatric cancers (Gröbner et al., Worst et al.)^{7,8} or neuroblastoma (Pugh et al., Ackermann et al., Brady et. al)^{1,3,4}. This compilation of cancer-related genes now comprises 801 genes and was added to the revised submission as Supplementary File 1. All affected figures have been changed accordingly. We agree that relevance for neuroblastoma has only been shown for a limited number of gene mutations. We changed Fig. 2a accordingly and marked documented neuroblastoma-related mutations in red (based on the aforementioned publications on neuroblastoma genetics). We also included Supplementary Table 2, which provides information about the type and putative biological consequences as well as expression of all the cancer-related SNVs of Fig. 2a.

2. The heatmaps in Fig. 2 are difficult to read and interpret due to the black background and the counter-intuitive use of lighter colors to show higher VAF. Inverting the color scheme would likely make these easier to interpret. For selected variants of special interest, such as ALK, FGFR1, etc., it would be helpful to show a bar or line plot with VAF on one axis and sample on the other, so that the VAFs can be assessed numerically rather than by color.

Response: We very much appreciate this comment from the reviewer and refer the reviewer also to our responses to reviewers #1 and #3, who made similar suggestions. We have thoroughly changed the coloring in the heatmaps in Fig. 2 and Supplementary Fig. 2a-h. We have switched the colors representing the VAF (now: high VAF as dark blue, low VAF as light blue on a white background) to display data in the heatmaps in a more intuitive, clearer, and easily interpretable way. We also added a bar plot displaying the VAF of selected cancer-related genes for the 2 patients in Fig. 2b and c.

3. Fig. 3 shows a valuable "average" summary across the cohort. It would also be helpful to show a few heatmaps of copy profiles, where each heatmap shows a specific patient having copy number heterogeneity. This would make it easier to evaluate and interpret the reported copy number heterogeneity within a specific patient. Panel B can be moved to Supplementary

as it is depicting a well-known knowledge that whole-genome duplication is more common in low-risk patients.

Response: We very much thank the reviewer for these comments. We performed an additional SCNA analysis using an updated version of the MEDICC software, which allows us to depict distinct SCNAs per patient in addition to phylogenetic sample trees. These results are shown in the new Supplementary Figure 4. We agree with the reviewer that Panel B only confirms common knowledge for neuroblastoma. Since Fig. 3 is a small figure, we left it there though.

4. *Distinct evolution trajectory of metastatic clone presented in Figure 4 is very interesting; however, the authors reached conclusion that metastatic clones were originated from early evolution branches (stated in abstract and results) based on the following data: “Both SCNA and SNV data indicated early branching of the metastatic clone, with clear evidence of clonal aberrations within the metastasis that were not present in the primary tumour (Fig. 4b, Supplementary Fig. 2a).” It should be noted that the argument for early evolution can only be inferred when the clonal variants in primary tumors are absent in metastasis but not the other way around. It will make more sense to place metastatic sample in parallel with the resected samples or as “descendants” of the diagnostic samples in Figure 4b, e. This also raised the question on the criteria used for constructing evolutionary tree using SCNA data.*

Response: We thank the reviewer for these comments. We agree that the phrasing was unfortunate, and we updated the paragraph accordingly. Indeed, the main argument for “early” branching of the metastasis in CB1001 is a region with LOH on chromosome 6q, which is present in the primary tumor, but absent in the metastasis. Since lost genetic material cannot be re-gained, the metastasis must have branched out before clonal diversification in the other samples in CB1001. The case is not so clear in CB1003, therefore, we reanalyzed all SCNA data with an improved version of our phylogeny reconstruction method, MEDICC2⁶. While all other trees remained as before, MEDICC2 placed the metastatic branch of CB1003 between the first and second look samples, in line with the gradual gain of *MYCN* copies reported here. We revised Fig. 4 to make this clearer, including marking up the defining synapomorphies of the clades and more clearly compare SNV-based to SCNA-based trees. Jackknife analysis of the trees using MEDICC2 also verified the robustness of the inferred topologies. We also revised Supplementary Fig. 4, which systematically compares SNV- and SCNA-based trees, to provide additional evidence about the robustness of the inferred tree topologies. For a more systematic comparison between SNV- and SCNA-based trees in other cancer types as well as between trees inferred by MEDICC or other methods please see for example^{6,12}.

5. *Does the UHR classification heterogeneity section (Fig. 5b) rely entirely on the fact that patients 1003 and 1008 had heterogeneity in MYCN and ALK alterations? The authors need to comment on that.*

Response: We thank the reviewer for pointing this out. The acquisition of the *MYCN* amplification in CB1003 and loss of the *ALK* R1275Q mutation in CB1008 are the most relevant probable causes for the heterogeneous UHR classification over the course of

disease. Tumor resection samples from CB1008 show that not only was the *ALK* mutation lost, but that *TERT* expression decreased substantially, while the *MYCN* amplification was present in both the biopsy at diagnosis and at tumor resection. This comment is now added in lines 345 to 347 to the revised manuscript.

6. *Figure 4b and e shown clonal structure derived from SCN analysis. It might be good to combine this together with the relevant graph in 4a so that it is clear the SCN analysis is consistent with those from SNV/indel. Supplementary Fig. 3 should also be modified to mark the CNV status integrated with SNV/indel. Figure 4f: sample 21 (metastatic sample) has the same logR profile (for CNV amplitude) as sample 23 (resected) but lacks the allelic imbalance shown in sample 23. It should be noted that according to text on page 11, sample 23 should have 1-copy gain ("all samples obtained from the resected tumour showed clear allelic imbalance and a single-copy gain of chromosome arm 9q, which was absent in both the diagnostic biopsies and the metastasis resected synchronously with the primary tumour (Fig. 4g)"). Figure 4b shown that sample 21 (metastatic sample) has the same amplitude of CNV (logR graph, 1-copy change) as sample 23 (resected sample), indicating a potential error in Figure 4d regarding the logR ratio of sample 21. The authors need to verify their data and confirm the copy number variation status of sample 21. If sample 21 is indeed having a logR comparable to sample 23 and lacks allelic imbalance, this can represent a subclonal two-copy gain (duplication of both haplotypes) and should be noted in the text or figure legend.*

Response: We are very grateful to the reviewer for spotting this discrepancy in the logR profile. We verified the copy number calls and realized this was an issue caused by assembling different parts of the logR and BAF tracks for the whole genome into one figure, where the y-axis scale was not adjusted. We apologize for this mistake and have fixed the issue and re-worked Fig. 4 to convey its message more clearly. The MSAI gain in CB1001 is now clearly visible in both logR and BAF profiles. Similarly, the 9q gain in CB1003 is now also supported by an increase in logR profile in sample 23 relative to profiles in samples 16 and 21.

7. *In the paper, two different evolution schemes are used: (a) the "ubiquitous/shared/private" scheme and (b) the "clonal/subclonal" scheme. Since these two schemes convey closely related information, it would be helpful to clarify to readers why both schemes were discussed, or alternatively, to use only one or the other. The criteria for defining clonal appears to be better suited for the definition of ubiquitous as it only requires a VAF of 0.1 (i.e. "SNVs were considered to be clonal if present with a variant allele frequency (VAF) greater than 10% in WES or targeted sequencing data from all samples. All other detected SNVs were considered subclonal").*

Response: We thank the reviewer for pointing this out. We harmonized the terminology in the revised manuscript and used the scheme clonal/subclonal according to the aforementioned definition of VAF (>10% and presence of the SNV in all sample from a patient). We chose this definition because it is also suitable for the description of clonal evolution and the phylogenetic trees.

8. *In patient CB1008, was the loss of the ALK variant after treatment due to a different anatomical site being sampled? Or was the same anatomical location sequenced in both the*

pre-treatment and post-treatment samples? This would indicate whether the change was related to temporal vs. spatial evolution. In general, indicating the specific site of each sample (e.g. adrenal, paraspinal, distant met location, etc.) in, for example, Fig. 1, would be useful to evaluate spatial vs. temporal differences.

Response: We thank the reviewer for this important question. The loss of the *ALK* mutation in CB1008 was detected in resection samples of the adrenal primary tumor. Sampling for the initial biopsies, which harbored the *ALK* mutation, was carried out before treatment from the same location. In all 3 cases, for which samples from two time points were included for WES analysis, the matched samples originated from the same anatomical site or the sampling site was indicated as (local) infiltration or (distant) metastasis. Fig. 1 and Fig. 2a have been changed accordingly.

9. Page 7 *“Spatial and temporal heterogeneity of single-nucleotide variants affects therapeutically actionable genes in neuroblastoma”*. Whereas between two and six non-synonymous SNVs were detected in known cancer-related genes (defined by the COSMIC database²⁰ and other studies^{6,10,21}) in high-risk tumours, very few or no non-synonymous SNVs were detected in neuroblastomas from other risk groups (Fig. 2a).

Is this related to the overall higher mutational burden in high-risk samples compared with the low risk samples.

Response: We thank the reviewer for this comment. Indeed, the overall higher mutational burden of the high-risk samples was true for all non-synonymous SNVs as well as the SNVs in cancer-related genes. This information is shown in Fig. 1 and mentioned in line 137. It is now also more precisely addressed to cancer-related genes in line 139.

Minor concerns

1. *The paper is well-written overall, but a few statements were unclear. For example, the statement “Focal amplifications of the MYCN oncogene, re-arrangements in the nearby TERT locus on chromosome 5p...” was confusing since MYCN and TERT are not nearby, but on different chromosomes. Additionally, the statement “On average, between 13-92% (median 34%, mean 42%) of each genome was affected by at least one SCNA event in at least one sample (Fig. 3a)” was a bit hard for me to follow. Does “each genome” refer to “each sample”? The terms “diagnosis of relapse” in the text and “relapse diagnosis” in Table 1 are also a bit ambiguous.*

Response: Thank you for these comments. We changed the text to assign the *TERT* rearrangement within the *TERT* locus in line 77. We revised line 195 from “each genome” to “Across the cohort, between 13-92% of the genome of each patient’s cancer was affected by at least one SCNA in at least one sample” and hope this makes it clearer. We also harmonized terminology to “diagnosis of relapse” in Table 1.

2. *Figure 2C/B: status of WES, FFPE or PDX can be combined into one bar to save space. Yellow grid plus the red lines makes it hard to discern the blue shade used for depicting the VAF. BRCA2 appeared twice in the same sample. Is this an error in data presentation?*

Response: Thank you for these comments. We changed the figure according to your suggestions and comments from reviewers #1 and #2. We also changed the background color of the heatmaps in Fig. 2 and inverted the color code representing the VAF for better visibility. We also added a bar plot depicting the VAF of the cancer-related genes in samples from patients CB1002 and CB1003 in Fig. 2b+c following the suggestions of reviewers #1 and #2. The BRCA2 SNVs appeared twice because they refer to adjacent, but different, SNVs in the regulatory region of the gene. All SNV positions and their potential biological consequences are listed in the revised Supplementary Table 2.

3. *Gains of 2p and 7q were observed in all or almost all of the 10 patients reported. This is higher than expected, based on the authors' references 6 and 16, for example. Is there some selection criteria for this study that might explain this?*

Response: We thank the reviewer for this comment. In the studies mentioned by the reviewer, gains in 2p and 7q are the predominant copy number gains and associated with poor overall survival in their cohort of >700 and 240 neuroblastomas, respectively^{4, 3}. Our relatively small cohort of 10 patients includes 7 high-risk patients and might, therefore, overrepresent these aberrations. This is in line with previous findings that a segmental chromosomal gain in 7q is more prevalent in high-risk neuroblastoma¹³.

4. *The trees in Fig. 4a and 4b were difficult for me to interpret. In particular, two numbering schemes are used, with (1) numbers inside the circles at the end of tree branches, and (2) a separate numbering system with (samples?) numbered in colors based on diagnosis/resection/relapse/metastasis status. Do these different numbering systems refer to mutation clusters vs. samples, respectively? Are these clone trees or sample trees? It's difficult to tell from the legend and clarification would be helpful.*

Response: We very much appreciate the reviewer comment and apologize for not being clear enough. We have thoroughly revised the manuscript text and added more detailed information to the results in Fig. 4 (lines 240 to 242). We now provide the explanation about (i) the numbers in circles that represent the clones and subclones and (ii) the numbers at the end of the branches, which represent the samples, to better enable interpretation of the data and SNV-based clone trees. Indeed, SNV-based trees are clone trees, where samples can occur multiple times within a tree, and SCNA-based trees are based on the dominant or major clone per sample. We have revised Fig. 4 to make this clearer, and now provide an extended systematic comparison (Supplementary Fig. 4) between SNV- and SCNA-based trees, showing very good overall agreement.

5. *From Supplementary Fig. 1a, it appears that more samples were subjected to targeted sequencing than WES. Were both WES and targeted sequencing performed on some samples, while other samples received targeted sequencing only? If yes, wouldn't the samples with targeted sequencing only lack any private variants since de novo detection of variants wasn't performed in these samples? Would this affect or confound any of the tree structures shown in Figs. 4a, b (i.e. were the targeted sequencing-only samples without WES included in these figures)?*

Response: We thank the reviewer for these important comments. Please also refer to our answer to reviewer #1 who has also raised similar questions. We made several

changes in the revised manuscript to better track patient samples and subsequent analyses thereof. We clarified in lines 108 to 110 that we performed targeted sequencing using a panel design based on 1476 SNVs detected by WES in 51 snap-frozen samples. Following WES analysis of the 51 samples, we included further 89 samples, to total 140 samples analyzed by ultra-deep targeted sequencing. We changed Fig. 2 accordingly. Yes, some samples were only analyzed by targeted sequencing, and these data were included in the trees in Fig. 4. In the panel sequencing, all samples were analyzed for selected SNVs pre-detected by WES in the same sample, but also for SNVs detected in all 51 samples from all patients. For this reason, some samples only analyzed by targeted sequencing could harbor private mutations (one example: sample 23 from patient CB1008).

6. Does the transcriptional change reported in CB1003 correspond to a shift between the two neuroblastoma transcriptional differentiation states reported in van Groningen et al., Nature Genetics 2017? If not, what specific transcriptional differences are seen before vs. after treatment? The transcriptional evolution of this patient, and the transcriptional differences between high- and low-risk patients, are of interest and it would be helpful to clarify what specific genes or gene signatures are variable in these settings.

Response: Thank you for pointing out these important comments. We refer the reviewer also to our responses to reviewer #1, point 4 and reviewer #2, major comments a) to d). We performed an analysis using gene signature scores (from ²) for patient samples from cases CB1003 and CB1008, but found no significant change in differentiation state signatures between two time points. We include this analysis as Supplementary Fig. 6f. This analysis might be limited by bulk RNA-Seq data and would be more informative using single-cell RNA-seq data, which could discriminate between tumor and stromal cells. The most striking temporal differences in transcript evolution in samples from patient CB1003 were as a consequence of the *MYCN* amplification that occurred later during tumor progression. GSEA confirmed this transcriptional outcome of the copy number changes at the *MYCN* locus. These analyses are now added to Supplementary Fig. 6. We also analyzed the transcriptional profile of high-risk versus intermediate- and low-risk tumors, but with inconclusive results. This was probably due to heterogeneity of the pre-treated or treatment-naïve samples, and the fact that only 3 patients (7 samples) belonged to either the intermediate- or low-risk groups.

7. The Supplementary Figure 2 text to indicate panels “f” and “h” appear to be missing.

Response: Thank you for this comment. We inserted the panel annotation in the revised Supplementary Fig. 2.

8. Figure 5a needs to mark the low- & high-risk tumor groups

Response: Thank you for this comment. We marked low- & high-risk tumor groups in the revised Fig. 5a.

References

1. Ackermann, S. *et al.* A mechanistic classification of clinical phenotypes in neuroblastoma. *Science (New York, N.Y.)* **362**, 1165-1170 (2018).
2. van Groningen, T. *et al.* Neuroblastoma is composed of two super-enhancer-associated differentiation states. *Nat Genet* **49**, 1261-1266 (2017).
3. Pugh, T.J. *et al.* The genetic landscape of high-risk neuroblastoma. *Nature Genetics* **45**, 279-284 (2013).
4. Brady, S.W. *et al.* Pan-neuroblastoma analysis reveals age- and signature-associated driver alterations. *Nat Commun* **11**, 5183 (2020).
5. Watkins, T.B.K. *et al.* Pervasive chromosomal instability and karyotype order in tumour evolution. *Nature* (2020).
6. Petkovic, M. *et al.* Whole-genome doubling-aware copy number phylogenies for cancer evolution with MEDICC2. *bioRxiv*, 2021.02.28.433227 (2021).
7. Gröbner, S.N. *et al.* The landscape of genomic alterations across childhood cancers. *Nature* **555**, 321-327 (2018).
8. Worst, B.C. *et al.* Next-generation personalised medicine for high-risk paediatric cancer patients - The INFORM pilot study. *Eur J Cancer* **65**, 91-101 (2016).
9. Schulte, J.H. & Eggert, A. ALK Inhibitors in Neuroblastoma: A Sprint from Bench to Bedside. *Clin Cancer Res* (2021).
10. Foster, J.H. *et al.* Activity of Crizotinib in Patients with ALK-Aberrant Relapsed/Refractory Neuroblastoma: A Children's Oncology Group Study (ADVL0912). *Clin Cancer Res* (2021).
11. Andersson, N. *et al.* Extensive Clonal Branching Shapes the Evolutionary History of High-Risk Pediatric Cancers. *Cancer Res* **80**, 1512-1523 (2020).
12. Schwarz, R.F. *et al.* Phylogenetic quantification of intra-tumour heterogeneity. *PLoS computational biology* **10**, e1003535 (2014).
13. Stallings, R.L. *et al.* Are gains of chromosomal regions 7q and 11p important abnormalities in neuroblastoma? *Cancer Genetics and Cytogenetics* **140**, 133-137 (2003).

REVIEWERS' COMMENTS

Reviewer #1 (Remarks to the Author):

The authors have adequately addressed all of my prior critiques and present a much improved manuscript that will be of interest to the cancer genomics and pediatric cancer communities.

Reviewer #2 (Remarks to the Author):

The authors have done an excellent job addressing my comments and adapted the manuscript accordingly.

Daniel Bexell

Reviewer #3 (Remarks to the Author):

Schmelz and colleagues have effectively addressed most of my concerns. In particular, the Figure 2 color scheme is much more readable now, and the annotation of credentialed neuroblastoma driver genes is helpful. The Figure 4 clone trees are also much clearer now, given the helpful text description of the numbering schemes. There are several minor concerns:

1. The updated wording regarding metastasis is difficult to interpret. For example, lines 38-41 in the abstract state "The genetic heterogeneity...support emergence of metastatic clones...during disease evolution". Likewise, lines 290-292 state: "Thus, SCNA analysis indicates that in a subset of neuroblastomas, metastases emerge during disease progression..." Both of these statements are very general and do not require genomic analysis for support. Perhaps the authors could replace the latter statement with something like, "the high genomic divergence between primary tumors and metastases suggests that in neuroblastoma, metastases spread from the primary tumor site early in disease evolution, although additional studies are required to clarify this." The statement in the abstract also needs some adjustments to make it meaningful.

2. I appreciate the authors checking the expression of putative driver mutations as shown in the new Supplementary Table 2. From this it appears that NOTCH (and several other genes/mutations noted as being heterogeneous "therapeutically actionable genes") are in fact not expressed in neuroblastoma. Given this, I would suggest not discussing these genes in the section on

heterogeneous disease drivers, though including them in Fig. 2a is probably okay given the new red text.

3. A possible minor correction: The authors state on lines 315-318 that “A principal-component analysis (PCA) utilizing expression levels of the most variable genes confirmed that interpatient differences dominated over intratumour heterogeneity in gene expression (Supplementary Fig. 6e).” I believe this is actually referring to Supplementary Figure 6g?

REVIEWERS' COMMENTS

Reviewer #3 (Remarks to the Author):

Schmelz and colleagues have effectively addressed most of my concerns. In particular, the Figure 2 color scheme is much more readable now, and the annotation of credentialed neuroblastoma driver genes is helpful. The Figure 4 clone trees are also much clearer now, given the helpful text description of the numbering schemes.

There are several minor concerns:

1. The updated wording regarding metastasis is difficult to interpret. For example, lines 38-41 in the abstract state "The genetic heterogeneity...support emergence of metastatic clones...during disease evolution". Likewise, lines 290-292 state: "Thus, SCNA analysis indicates that in a subset of neuroblastomas, metastases emerge during disease progression..." Both of these statements are very general and do not require genomic analysis for support. Perhaps the authors could replace the latter statement with something like, "the high genomic divergence between primary tumors and metastases suggests that in neuroblastoma, metastases spread from the primary tumor site early in disease evolution, although additional studies are required to clarify this." The statement in the abstract also needs some adjustments to make it meaningful.

Response: We thank the reviewer for pointing this out and changed the text in line 291 to 293 according to the suggestion to "Thus, SCNA analysis indicates high genomic divergence between primary tumor and metastases in neuroblastoma suggesting that metastases spread early in disease evolution from the primary tumor site, although additional studies are required to further substantiate this. We also adjusted the abstract phrasing in line 40 to "early emergence of metastatic clones".

2. I appreciate the authors checking the expression of putative driver mutations as shown in the new Supplementary Table 2. From this it appears that NOTCH (and several other genes/mutations noted as being heterogeneous "therapeutically actionable genes") are in fact not expressed in neuroblastoma. Given this, I would suggest not discussing these genes in the section on heterogeneous disease drivers, though including them in Fig. 2a is probably okay given the new red text.

Response: We thank the reviewer for this comment and excluded *NOTCH2* and *CCND3* in the aforementioned section in line 145.

3. A possible minor correction: The authors state on lines 315-318 that "A principal-component analysis (PCA) utilizing expression levels of the most variable genes confirmed that interpatient differences dominated over intratumour heterogeneity in gene expression (Supplementary Fig. 6e)." I believe this is actually referring to Supplementary Figure 6g?

Response: We thank the reviewer for this comment and changed the figure reference in line 318 to Supplementary Fig. 6g.